# Transient inhibition of cell division in competent pneumococcal cells results from deceleration of the septal peptidoglycan complex

**Dimitri Juillot**[1,2,3], **Cyrille Billaudeau** [1,4], **Isabelle Mortier-Barrière**[2,3,4], **Aurélien Barbotin** [1], **Armand Lablaine**[1], **Patrice Polard** [2,3], **Nathalie Campo** [2,3] ✉ & **Rut Carballido-López** [1] ✉

Membrane protein ComM transiently inhibits cell division during the development of the competence state in the pathogenic bacterium *Streptococcus pneumoniae*, but the underlying molecular mechanisms remain unclear. Here we show that, in competent cells, ComM moves together with, and reduces the speed of, septal peptidoglycan synthetic complex FtsW:PBP2x. ComM directly interacts with the putative FtsW:PBP2x activator DivIB, and overproduction of DivIB counteracts FtsW:PBP2x deceleration along the cell division delay in competent cells. Our results support a model in which ComM reduces septal peptidoglycan synthesis by interfering with DivIB activity during competence in *S. pneumoniae*.

In bacteria, environmental and cellular stresses including nutrient exhaustion, envelope stress and DNA damages, trigger various signaling pathways that can cause a transient inhibition of the cell division process to halt the cell cycle progression and allow time for repair[1,2]. Inhibition of cell division is also frequently associated to differentiation programs such as sporulation in *Bacillus subtilis*[3], heterocysts formation in Anabaena[4,5] and aerial hyphal development in Streptomyces[6–8]. However, most cellular factors and mechanisms arresting cell division in response to specific conditions that enable bacteria to adapt remain largely unknown. The best characterized mechanism involves the SOS cell division inhibitor SulA in *Escherichia coli*, which is induced following activation of the RecA protein by damage to DNA[9]. SulA binds to monomers of the bacterial tubulin homolog FtsZ in the cytoplasm, and prevents their polymerization into a ring-like structure (the Z-ring) at the future division site[10]. The Z-ring marks the site of cell division and directs the assembly of the cell division machinery[11]. While other factors such as MciZ in *B. subtilis*[12] also impact the assembly or stability of the Z-ring, most known cell division inhibitors localize in the membrane and interfere with the functions of various components of the cell division machinery[13–19].

This machinery, also called the divisome[11,20], is a large multi-protein complex containing proteins involved in cell division and septal cell wall peptidoglycan (PG) synthesis that collectively achieve membrane invagination during cytokinesis and build a crosswall (the division septum), which will eventually split to produce two daughter cells. Among the divisome components, FtsZ plays a major role by coordinating the cell constriction process. FtsZ filaments have been shown to treadmill circumferentially around the division site in several bacterial species, with an average treadmilling speed of ~30 nm/s[21–25]. The link between the dynamics of FtsZ filaments and the activity of PG synthases has been addressed in several bacterial species, both Gram-positive and Gram-negative: *Bacillus subtilis*[26,27], *Streptococcus pneumoniae*[25] and *Staphylococcus aureus*[28], *Escherichia coli*[29–32], *Caulobacter crescentus*[33]. The model that has emerged is that FtsZ treadmilling drives Z-ring condensation and affects septum constriction to different extents in different bacteria, but does not control the

[1]Université Paris-Saclay, INRAE, AgroParisTech, Micalis Institute, 78350 Jouy-en-Josas, France. [2]Laboratoire de Microbiologie et Génétique Moléculaires, UMR5100, Centre de Biologie Intégrative, Centre Nationale de la Recherche Scientifique, 31062 Toulouse, France. [3]Université Paul Sabatier (Toulouse III), 31062 Toulouse, France. [4]These authors contributed equally: Cyrille Billaudeau, Isabelle Mortier-Barrière. ✉e-mail: nathalie.campo@univ-tlse3.fr; rut.carballido-lopez@inrae.fr

                                                        

processive movement of active septal PG synthases as originally suggested[21]. A common feature of all these systems is a direct link between the velocity of specific components of the divisome and the rate of PG synthesis and septum closure. Importantly, several cell division inhibitors blocking the constriction process have been shown to directly interact with the divisome in various bacteria[13–15,17], but the impact of these interactions on the dynamics of the targeted proteins remains unexplored.

Here, we investigated the mechanism of transient cell division inhibition observed during the development of competence for transformation in the human pathogen *S. pneumoniae* (the pneumococcus). Natural genetic transformation is a conserved mechanism of horizontal gene transfer, which enables the cells to acquire new genetic traits and repair DNA damages in bacteria[34]. It requires that cells enter a differentiation state called competence during which exogenous DNA is imported into the cell and integrated into the chromosome by homologous recombination. In *S. pneumoniae*, competence is transient. It relies on a secreted peptide pheromone known as the competence-stimulating peptide (CSP), that triggers competence in all cells in the population during exponential phase, and persists over a short period of time, shorter than the cell generation time (~25 min and ~30 min, respectively)[35–39]. Remarkably, the cell division process is delayed in competent pneumococcal cells, and evidences indicate that this delay contributes to stress tolerance in competent cells[40] and preservation of genomic integrity during transformation[38]. ComM, a membrane protein specifically induced during competence, appears necessary and sufficient to inhibit cell division and to interfere with PG synthesis[38]. ComM also confers immunity to a killing mechanism termed fratricide that can be used by competent cells to acquire DNA from non-competent pneumococci[41–43]. ComM localizes at midcell and does not affect assembly and localization of divisome components, suggesting that it transiently inhibits the active process of constriction[38]. In *S. pneumoniae*, both septal and peripheral (elongation) PG synthesis localize at the site of cell division and occur concomitantly throughout the cell cycle[25,44]. Septal PG synthesis is mediated by the septal FtsW transglycosylase and its PBP2x transpeptidase binding partner, which move together around septal rings[25,44,45], presumably activated by the DivIB/DivIC/FtsL complex[28,46,47]. Concomitantly, the RodA transgycosylase and its cognate PBP2b transpeptidase mediate peripheral PG synthesis and thus elongation of the ovococcal cell[44,45,48,49].

In this study, we sought to investigate the molecular mechanism underlying the ComM-mediated cell division delay of competent cells. We show that ComM is a dynamic protein that co-localizes with and slows the speed of the FtsW:PBP2x complex in *S. pneumoniae*. Furthermore, we provide evidence suggesting that ComM may reduce septal PG synthesis by interacting with DivIB, thereby preventing the activation of the septal PG synthase complex.

## Results

### ComM moves processively around the division site of competent cells

To investigate how ComM interferes with the cell division process during competence, we first examined its dynamics at the division site. For this, we engineered a strain expressing a functional mNeonGreen-ComM fusion from the native *comM* chromosomal locus, under the control of its native competence-induced promoter. Note that all strains used in this study were engineered in the *ΔcomC* background, which renders them unable to develop competence spontaneously[50], and that competence is then induced by addition of synthetic CSP[35]. The mNeonGreen-ComM fusion was produced specifically in competent cells with no detectable proteolysis (Supplementary Fig. 1a). Protein levels reached a maximum about 15 min after competence induction and then progressively decreased (Supplementary Fig. 1b). The functionality of the fusion was assessed as per its ability to delay

cell division and lead to cell elongation during competence[38,39] (Supplementary Fig. 2). Epifluorescence microscopy of competent cells lying horizontally on the slide confirmed that mNeonGreen-ComM localizes as a band at the site of division (Supplementary Fig. 3), as expected[38].

To resolve the localization pattern of ComM at the division site with optimal resolution, we vertically immobilized pneumococcal cells in agarose microholes at 37 °C and used highly inclined laminated optical sheet (HILO)[51] microscopy to visualize the entire division plane (Fig. 1a). This revealed that ComM forms several independent patches that move around the cell circumference (Fig. 1b, c and Supplementary Movie 1). Total internal reflection fluorescence microscopy (TIRFM) of cells lying horizontally (Fig. 1d) confirmed directional movement of ComM at the division site (Fig. 1e–g and Supplementary Movie 2). ComM patches moved in both directions around the cell circumference, with an average speed of 12.2 ± 5.3 nm·s$^{-1}$ (Fig. 1g, h). The same speed was measured independently in vertically oriented cells from kymographs generated around the cell circumference (Supplementary Fig. 4). Notably, ComM speed was steady throughout the development of competence, which lasts for about ~25 min within the population[35,37], and remained constant afterward (Fig. 1i). However, the number of cells displaying a septal ComM signal progressively decreased starting 20 min after competence induction (Supplementary Fig. 1c).

### Dynamics of FtsZ is unaffected during competence

We previously showed that in competent cells, Z-rings properly assemble at the division site, but the initiation and completion of the constriction process are transiently inhibited[38]. FtsZ treadmilling around the division site is essential for cell division[21]. It was found to be required for the initiation of septum constriction, after which it becomes dispensable for division in both *S. aureus*[23] and *B. subtilis*[26]. However, septum synthesis (constriction) requires the movement of septal PG synthases, which is independent of FtsZ treadmilling in both *S. aureus*[28] and *S. pneumoniae*[25], while in *B. subtilis*, it appears to be partially dependent on it[27]. To test whether competence interferes with FtsZ activity in *S. pneumoniae*, we compared the dynamics of FtsZ in competent and non-competent cells. Cells expressing from the native *ftsZ* locus a functional FtsZ-mNeonGreen fusion (Supplementary Fig. 1a, 3, 5) were grown to early exponential phase and induced (or not) to develop competence. Demographs displaying fluorescence intensity as a function of cell length showed that the deployment of the Z-rings during the cell cycle, from the septa of early divisional cells to the equators of the daughter cells, is unaffected during competence (Supplementary Fig. 6a). Maximum projections of TIRFM time-lapses yielded typical patterns of pre-divisional and dividing pneumococcal cells containing mature, equatorial and nascent Z-rings (Fig. 2a), as previously described by the Winkler laboratory[25]. FtsZ rings of early divisional cells become mature septal rings at the onset of division, when most divisome proteins are recruited; then nascent rings form on either side of the mature septal ring and start to move outward to develop into equatorial rings in the future daughter cells at the offset of division (Fig. 2b)[25]. Analysis of the time-lapses confirmed bidirectional movements within the Z-rings (Supplementary Fig. 7a–c, Supplementary Movie 3), with an average FtsZ filament velocity of about 30 nm·s$^{-1}$ (Fig. 2c), in line with earlier reports in *S. pneumoniae*[25] and other bacteria[21,22]. Importantly, FtsZ velocity remained unaffected during competence (Fig. 2c, d). We wondered whether the structure of the Z-rings or the rate of FtsZ treadmilling could be nevertheless impacted at specific stages of the cell cycle, which might have been masked in our ensemble analysis of the Z-ring population. Precise measurements of the density (total Z-ring intensity) and the velocity of FtsZ filaments within five distinct categories of Z-rings with different degrees of maturity over six different stages of the cell cycle produced similar values both in non-competent and competent cells at either

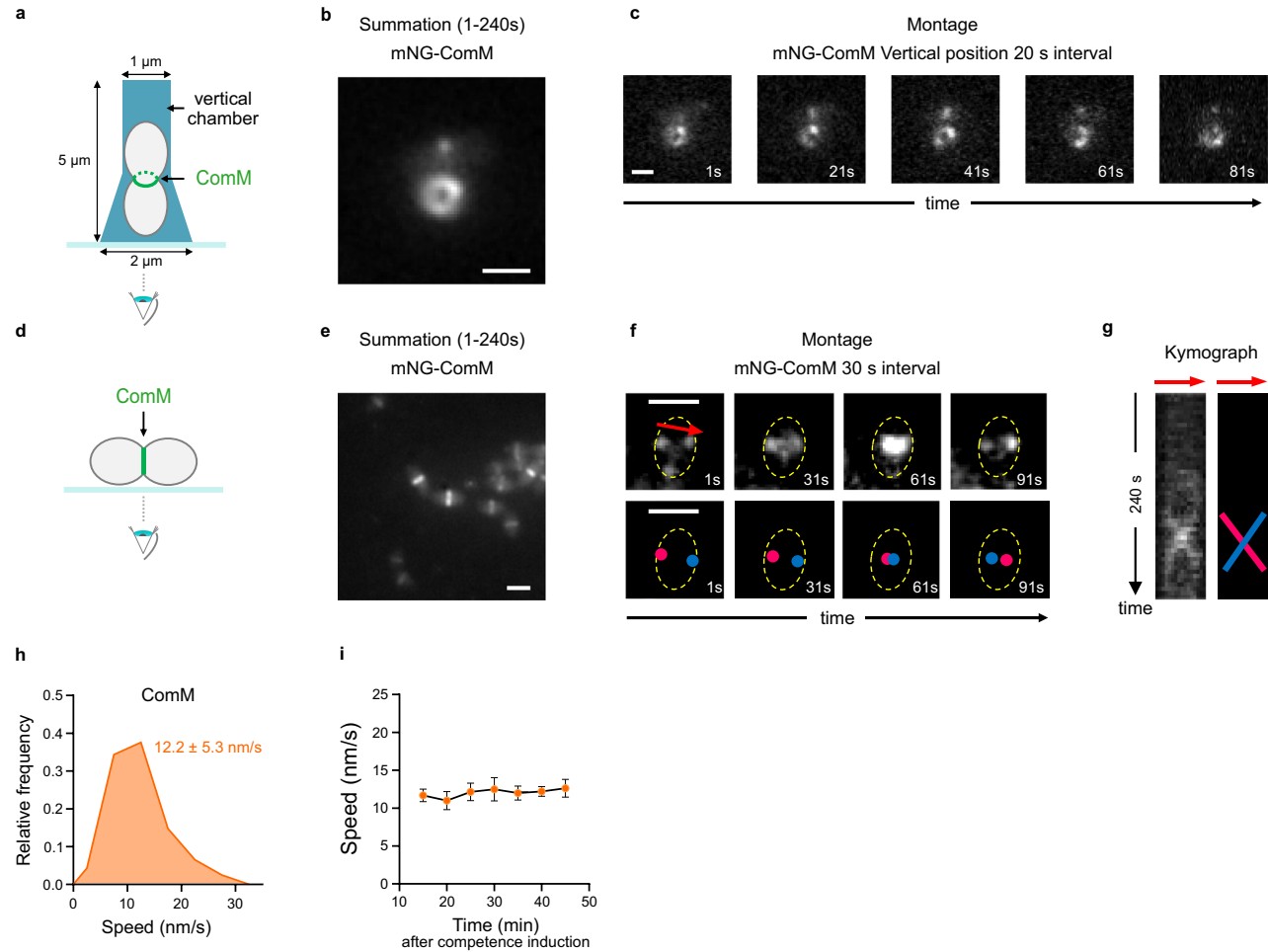

**Fig. 1 | ComM moves processively around the division site.** Cells expressing mNeonGreen-ComM (strain R4601) were grown in C + Y medium to early exponential phase and induced to develop competence by CSP addition just before imaging. Scale bars, 1 μm. **a** Schematics of the method to observe *S. pneumoniae* cells trapped in vertical chambers with the division plane parallel to the coverslip (related to panels b and c). **b** Representative fluorescence intensity projection. Summation of frames from 240 s HILO movies of mNeonGreen-ComM (3 s intervals). Images representative of three independent biological replicates. **c** Montage of images at 20 s intervals showing mNeonGreen-ComM in vertically oriented cells. **d** Schematics of the method to observe *S. pneumoniae* lying horizontally with the division plane orthogonal to the coverslip (related to panels e and f). **e** Summation of frames from 240 s TIRFM movies of mNeonGreen-ComM (3 s intervals). Images representative of three independent biological replicates. **f** Top panels, montage of

TIRFM images at 30 s intervals showing mNeonGreen-ComM in cells lying horizontally. The red arrow indicates the trajectory extracted for kymograph analysis (voluntarily slightly shifted to the top to allow the visualization of ComM patches). Bottom panels, cartoon representation of the two ComM patches (red and blue dots). The cell contour is represented in yellow dotted lines. **g** Kymograph from 1–240 s obtained from the ComM trajectory shown in panel f (left), and its cartoon representation (right). The slope of the two ComM patches shown in panel f are displayed. **h** Distribution of speed of mNeonGreen-ComM in competent cells (*n* = 812 trajectories analyzed over three independent biological replicates). Average speed ± sd is indicated. **i** mNeonGreen-ComM average speed ± sd measured at different time points after competence induction (same dataset as panel h). Error bars represents 95% confidence interval. Source data are provided as a Source data file.

early or late times of competence (Fig. 2b, e, f and Supplementary Fig. 7d). Altogether, these results demonstrate that the localization and treadmilling of FtsZ remain unaffected during competence in *S. pneumoniae*.

## The septal PG synthase complex FtsW:PBP2x decelerates in competent cells, while the speed of the peripheral PG synthase complex RodA:PBP2b remains unaffected

As previous labeling experiments of PG synthesis indicated that the rate of PG incorporation at the division site is reduced during competence[38], we next examined the dynamics of components of the PG synthetic machineries in competent cells. In *S. pneumoniae*, both septal and peripheral PG syntheses, catalyzed by the SEDS-family transglycosylase:bPBP transpeptidase pairs FtsW:PBP2x and the RodA:PBP2b, respectively, occur at the site of cell division and progress separately[44,45]. We hypothesized that the competence-dependent cell division delay might result from a defect of the septal

PG machinery. FtsW and its partner PBP2x have been shown to move along septal rings at the same speed, which is slower than FtsZ treadmilling speed and depends of PG synthesis[25]. We constructed functional fusions of FtsW and PBP2x to the mNeonGreen fluorescent protein expressed from the native chromosomal loci. The fusions were expressed at comparable levels and did not alter growth or cell morphology in either competent and non-competent cells, and showed the typical localization at mature and equatorial rings (Fig. 3a and Supplementary Figs. 1a, 3, 5). Demographs indicated that the two proteins relocate to the equators of future daughter cells late in the cell cycle, after FtsZ deployment, in both non-competent[25] and competent cells indistinctly (Supplementary Fig. 6). In non-competent cells, the mNeonGreen-PBP2x and FtsW-mNeonGreen fusions exhibited bidirectional processive movements at the site of cell division, with average speeds of 19.8 ± 6.2 nm·s⁻¹ and 19.8 ± 6.9 nm·s⁻¹, respectively (Fig. 3a, b and Supplementary Fig. 8, Supplementary Movies 4, 5), similar to values reported earlier

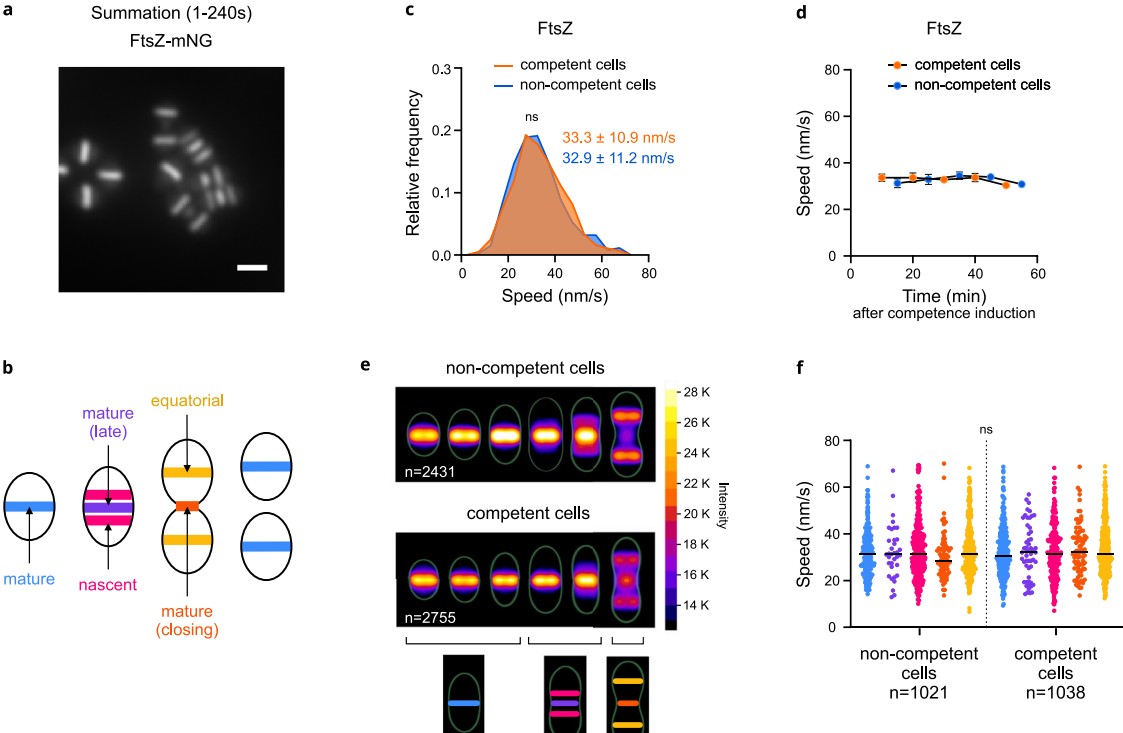

**Fig. 2 | Dynamic of FtsZ is unaffected during competence.** Cells expressing FtsZ-mNeonGreen (strain R4599) were grown in C + Y medium to early exponential phase and competence was induced, or not, by CSP addition. **a** Representative fluorescence intensity projection. Summation of frames from 240 s TIRFM movies (3 s intervals). Scale bar, 1 μm. Images representative of two independent biological replicates. **b** Schematic representation of pneumococcal cells in their cell cycle. Five distinct type of Z-rings are represented. **c** Distribution of speed of FtsZ-mNeonGreen in competent and non-competent cells (566 and 878 trajectories, respectively, measured over two independent biological replicates). Average speed ± sd for each condition is indicated. ns, no significant difference according to the two-sided Mann–Whitney nonparametric test (-*P* = 0.262). **d** FtsZ-mNeonGreen average speed ± sd at different time points after CSP addition (competent cells, 922

trajectories) or not (non-competent cells, 978 trajectories). Error bars represent 95% confidence interval. **e** Heatmaps representing the average fluorescence intensity and localization pattern of FtsZ-mNeonGreen in cells arranged according to size, in non-competent (top) and in competent (middle) cells, and cartoon representation of the corresponding Z-ring types for each cell categories (bottom). *n* values represent the number of cells analyzed in a single representative experiment. **f** FtsZ speed recorded in the different Z-ring types (same color code as in b) in non-competent and competent cells (1021 and 1038 trajectories (n) were analyzed, respectively, measured over two independent biological replicates). Two-sided one-way ANOVA statistical analysis showed no significant differences between conditions (ns, exact *P* = 0.3924). Source data are provided as a Source data file.

for Halo-tagged PBP2x and FtsW fusions[25]. In addition, FtsW and PBP2x exhibited diffusive behavior along the membrane (higher background signal in TIRFM summations and kymographs relative to FtsZ) (compare Supplementary Fig. 7 and 8), as previously reported too[25]. Such diffuse signal did not result from cleavage/degradation of the fluorescent fusions (Supplementary Fig. 1a). Importantly, the average speed of the two fusions was reduced in competent cells (Fig. 3b). We monitored their dynamics as a function of time after CSP addition and found that both mNeonGreen-PBP2x and FtsW-mNeonGreen slowed down to about 16 nm·s⁻¹ at 10 min after competence induction and then gradually recovered, to reach non-competent-like speed 40–50 min after competence induction (Fig. 3c). These results show that the development of competence causes a transient deceleration of PBP2x and FtsW motion.

We then wondered if the peripheral PG synthesis machinery was also affected during competence. Functional fusions of RodA to mNeonGreen (RodA-mNeonGreen) and of PBP2b to GFP (GFP-PBP2b)[52] expressed from the native loci (Supplementary Fig. 1a, 5) localized to the site of division (Fig. 3a and Supplementary Fig. 3) and displayed the same deployment pattern from mature to equatorial rings than FtsW and PBP2x in both competent and non-competent cells (Supplementary Fig. 6b, c). In non-competent cells, RodA and PBP2b also displayed bidirectional processive movement around the septal rings, albeit at slightly different speeds compared

to the FtsW:PBP2x complex, along with some diffusive behavior (Fig. 3b, Supplementary Fig. 8 and Supplementary Movies 6, 7), as recently reported[53]. Circumferentially moving RodA and PBP2b molecules are reflective of PG synthesis whereas molecules diffusing throughout the cell are presumably not synthesizing PG[53]. Importantly, the speed of the circumferentially moving population remained unchanged after CSP addition and throughout competence (Fig. 3b, c), suggesting that competence induction has no effect on the peripheral PG machinery. Unperturbed activity of the peripheral PG elongation machinery during competence, when septal PG synthesis is reduced, may explain the increased cell length of competent cells relative to non-competent cells[38].

Taken together, these findings show that the bulk of both the FtsW:PBP2x and RodA:PBP2b pairs populations exhibit circumferential motions along septal rings significantly slower than the speed of FtsZ treadmilling in either competent (Fig. 3d and Supplementary Table 1) and non-competent cells (compare Figs. 2c and 3b). However, their speeds are slightly different, suggesting that the septal and peripheral PG machineries do not move together[53]. This is also consistent with the finding that PBP2x and PBP2b localize into separate concentric rings as septation progresses[45]. Importantly, the speed of the septal FtsW:PBP2x complex but not of the peripheral RodA:PBP2b complex is slowed down during competence, suggesting that septal PG synthesis is reduced and may explain the delay of division in pneumococcal competent cells.

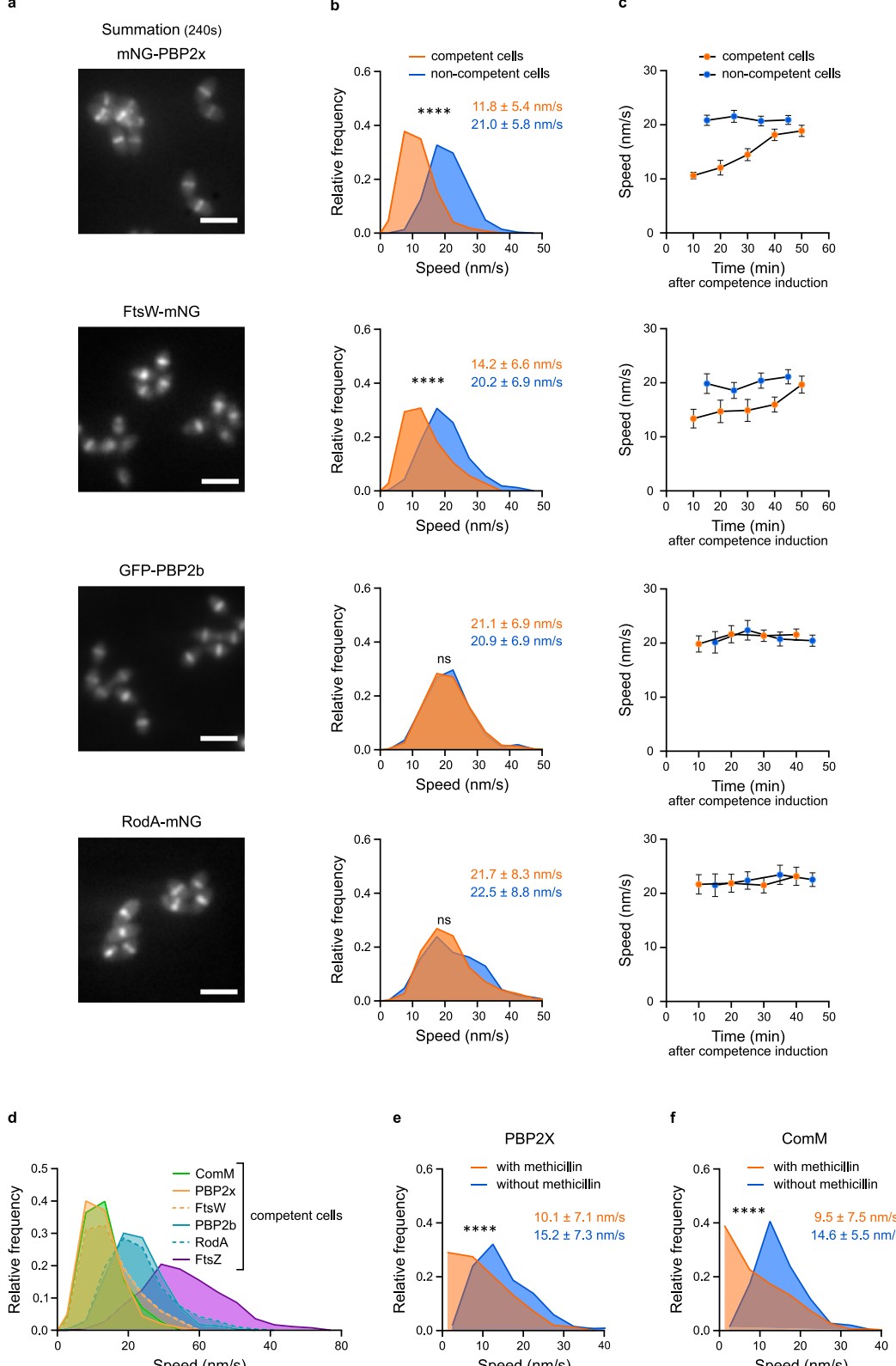

## ComM and PBP2x move together in competent cells

We immediately noticed that in competent cells, the FtsW:PBP2x pair slowed down to reach a mean speed similar to that of ComM (Fig. 3d and Supplementary Table 1), suggesting that they might move together along septal rings. FtsW:PBP2x processive movement has been shown to depend on PG synthesis in *S. pneumoniae*[25]. Septal PG synthesis, in turn, depends on PBP2x transpeptidase

activity[54–56], and PBP2x is required to stimulate FtsW transglycosylase activity[57]. In agreement with this, in the presence of the β-lactam methicillin at a concentration that mainly inhibits PBP2x transpeptidase activity[56,58], PBP2x motion is virtually halted in non-competent cells, while FtsZ motion remains unaffected (Supplementary Fig. 9)[25]. To test whether ComM speed is linked to FtsW:PBP2x speed, we tested the effect of methicillin in competent cells. The

**Fig. 3 | The speed of PBP2x and FtsW is reduced in competent cells to match that of ComM.** Cells were grown in C + Y medium to early exponential phase and induced to develop competence by CSP addition before imaging by time-lapse TIRFM. Strains analyzed: R4743 (mNeonGreen-PBP2x), R4728 (FtsW-mNeonGreen), WT gfp-pbp2b (GFP-PBP2b), R4867 (RodA-mNeonGreen), R4599 (FtsZ-mNeon-Green) and R4601 (mNeonGreen-ComM). **a** Representative fluorescence intensity projection images. Summation of frames from 240 s HILO movies at 3 s intervals. Scale bars, 2 μm. Images representative of two independent biological replicates. **b** Distribution of speed in competent and non-competent cells. Average speed ± sd is indicated. Number of trajectories measured over two independent biological replicates in non-competent and competent cells are respectively 615 and 363 for PBP2x, 323 and 143 for FtsW, 365 and 300 for PBP2b and 611 and 298 for RodA. Pairwise comparisons were done with a nonparametric two-sided Mann–Whitney test. ****, approximate P-values < 0.0001 were calculated for mNG-PBP2x and FtsW-mNG. ns, no significant difference for GFP-PBP2b (-P = 0.6176) and RodA-mNG (-P = 0.1295). **c** Average speed ± sd measured at different time points after competence induction. Error bars represent 95% confidence interval. Number of trajectories measured over two independent biological replicates in non-competent and competent cells are respectively 615 and 648 for PBP2x, 323 and 359 for FtsW, 365 and 447 for PBP2b and 475 and 394 for RodA. **d** Distribution of speed of FtsZ-mNeonGreen, mNeonGreen-PBP2x, FtsW-mNeonGreen, GFP-PBP2b, RodA-mNeonGreen and mNeonGreen-ComM patches in competent cells. Same data set as in Figs. 1h, 2c and 3b. **e** Distribution of speed of mNeonGreen-PBP2x patches in competent cells treated, or not, with methicillin (401 and 455 trajectories, respectively, measured over two independent biological replicates). Average speed ± sd for each condition is indicated. ****, approximate P-value < 0.0001 (nonparametric two-sided Mann–Whitney test). **f** Distribution of speed of mNeonGreen-ComM patches in competent cells treated, or not, with methicillin (190 and 298 trajectories, respectively, measured over two independent biological replicates). Average speed ± sd for each condition is indicated. ****, approximate P-value < 0.0001 (nonparametric two-sided Mann–Whitney test).

velocities of both PBP2x and ComM were drastically reduced in the presence of 0.3 μg ml⁻¹ methicillin (Fig. 3e, f), suggesting that they share a common motor. Additionally, the kinetics of deployment of ComM from mature to equatorial rings was similar to that of PBP2x and FtsW (Supplementary Fig. 6b, d). To further investigate this, we generated a strain co-expressing mNeonGreen-ComM and HaloTag-PBP2x (Supplementary Fig. 1a, 3, 5). Two-color demographs confirmed that the deployment of the two proteins is synchronized during the cell cycle (Fig. 4a). Altogether, these results indicate that ComM and FtsW:PBP2x exhibit the same dynamics in competent cells.

The lateral resolution of TIRFM ( ~ 250 nm) is not sufficient to assess the colocalization of independent patches of ComM and PBP2x at the division rings (Supplementary Fig. 10a). We then moved to super resolution microscopy using Lattice dual iterative structural illumination microscopy (SIM²), which allows imaging at high speed and doubles the conventional resolution of SIM[59]. We achieved localizations of mNeonGreen-ComM and HaloTag-PBP2x with measured spatial resolution of ~70 nm and ~80 nm, respectively, in SIM² images, compared to ~140 nm and ~160 nm in SIM images (Supplementary Fig. 10b, c). No flow through was detected between channels using strains that express the individual fusions (Supplementary Fig. 10c), confirming that these were suitable for colocalization experiments. Two-color 2D-SIM² of horizontally immobilized cells demonstrated extensive colocalization of the ComM and PBP2x fusions at the septal rings (Fig. 4b, c). Imaging the entire division plane in vertically immobilized cells by 2D-SIM² further showed that ComM and PBP2x form sparsely distributed patches that partially colocalize around the cell circumference (Fig. 4d and Supplementary Fig. 10d). In contrast, mNeonGreen-ComM and a functional RodA-HaloTag fusion (Supplementary Figs. 3, 5) showed more limited colocalization (Fig. 4e and Supplementary Fig. 10e). Pearson's Correlation Coefficient (PCC) analysis of the fluorescence signals in the two channels in lattice SIM images (prior to SIM² reconstruction, which may affect the linearity of the fluorescent signal) confirmed that ComM colocalizes more strongly with PBP2x than with RodA (Fig. 4g). The PBP2x ring (septal PG machinery) was previously shown to separate to the centers of septa (constricting inner ring) at the late stages of cell division whereas the adjacent PBP2b ring (peripheral PG machinery, containing RodA too[53]) remains peripheral to it (outer ring)[44,45,55]. Notably, ComM and PBP2x rings displayed the same diameter at different division stages (Fig. 4d, h, k), suggesting that the two proteins migrate together to the centers of constricting septa. However, the diameter of ComM rings was found essentially smaller than the diameter of the rings formed by RodA in the outer ring (Fig. 4e, i, k). Taken together, these data provide strong evidence that ComM and PBP2x coexist in motile patches at the septal ring of competent cells.

## ComM reduces PBP2x speed in both competent and non-competent cells

The rate of septal PG synthesis is thought to determine FtsW:PBP2x speed at the septal ring[25], and ComM was shown to play an important role in the reduction of the rate of PG synthesis in competent cells[38]. We then wondered if the reduction of speed of FtsW:PBP2x observed during competence depends on ComM. When competence was induced, a ΔcomM mutant expressing the mNeonGreen-PBP2x fusion did not show a delay in cell division (Supplementary Fig. 2a, b) as expected[38], and PBP2x speed did not decrease as observed in the wild-type background (Fig. 5a). We concluded that the reduction of speed of PBP2x observed in competent cells depends on ComM. To test if ComM is sufficient to slowdown PBP2x, we measured PBP2x speed in cells producing ComM outside competence. For this, we used a strain expressing comM from an ectopic locus under the control of an inducible promoter[38]. In growing cells, PBP2x displayed the same average speed as in wild-type cells in the absence of inducer (Fig. 5b, and Supplementary Table 1). However, in the presence of inducer, PBP2x speed was significantly reduced (Fig. 5b). Furthermore, in the presence of inducer the average doubling time and cell length of the population increased, mirroring these two ComM-dependent phenotypes observed in competent cells (Supplementary Fig. 2a, b)[38]. Taken together, these results indicate that ComM alone is sufficient to slowdown the septal PG synthetic complex.

## ComM interacts with the FtsW:PBP2x putative activator DivIB

We next sought to investigate the mechanism by which ComM reduces FtsW:PBP2x speed to delay cell division. Since FtsW:PBP2x speed depends on septal PG synthesis rate[25], we hypothesized that ComM may decrease it by reducing the activity of the complex. We found no interaction between the membrane proteins ComM and FtsW or PBP2x in yeast two hybrid (Y2H) assays (Fig. 6a and Supplementary Fig. 11a), suggesting that ComM does not affect FtsW:PBP2x activity through a direct protein-protein interaction. We then performed a genome-wide Y2H screen to search for potential interaction partners of ComM, used as bait, and identified a prey clone expressing a presumably cytoplasmic fragment of the bitopic membrane protein DivIB (aa 14-82 over 396 aa total length, Supplementary Fig. 11a). Specificity Y2H assays in which the cytoplasmic N-terminal (aa 1–130/396) and the extracellular C-terminal (aa 150–396/396) domains of DivIB were independently cloned confirmed the direct interaction between ComM and DivIB (Fig. 6a). Our Y2H assays also confirmed the direct interaction between DivIB and FtsW previously reported in B. subtilis[60] and S. pneumoniae[61] (Fig. 6a). To obtain evidence in support of the ComM-DivIB interaction in pneumococcal cells, we used a split-luciferase complementation assay[62], which relies on a luciferase variant separated into a large (LgBit) and small (SmBit) fragment. We constructed a strain producing both LgBit-ComM and SmBit-DivIB fusions from their native loci. As

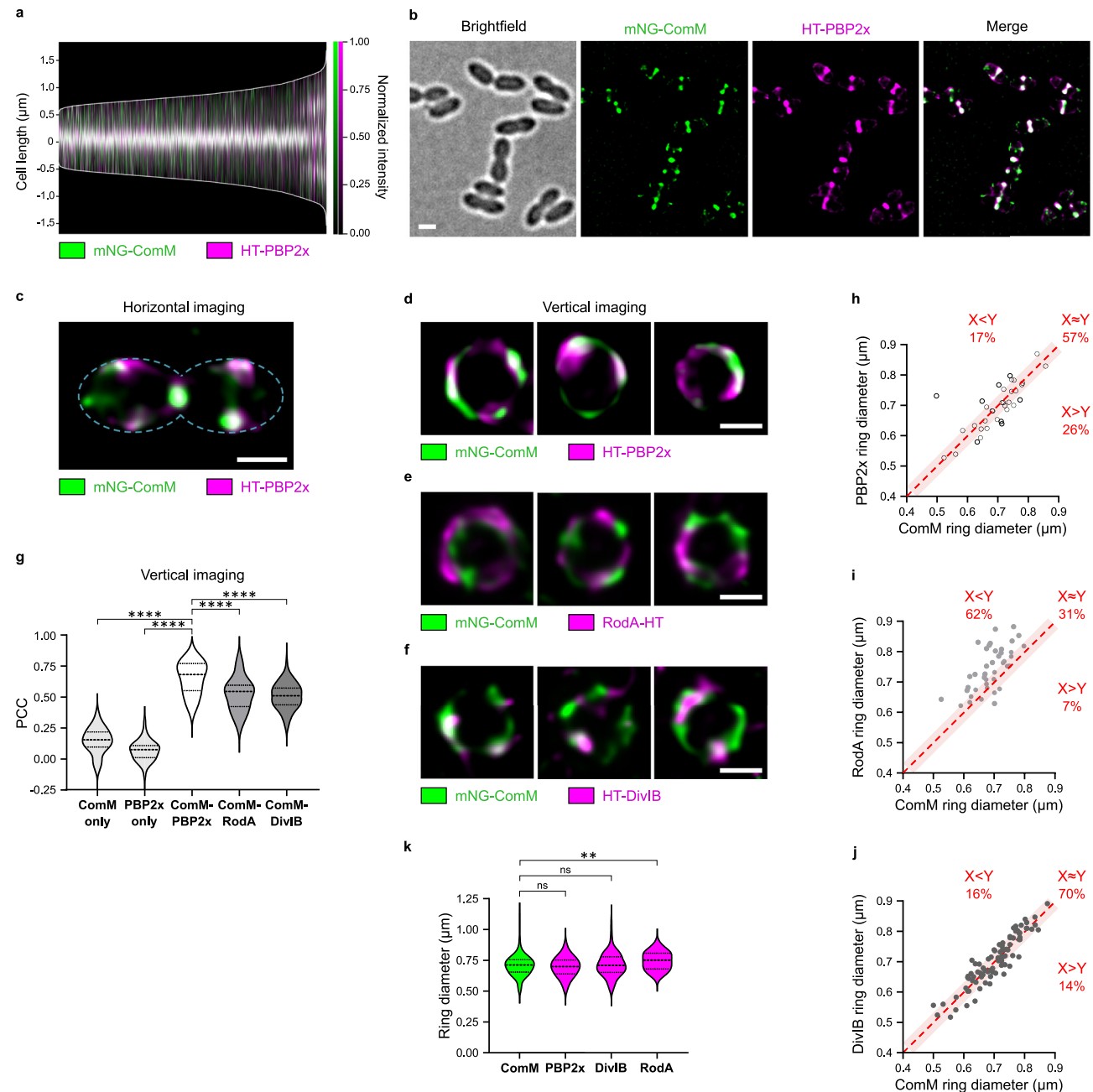

**Fig. 4 | ComM and PBP2x partially colocalize.** Strains R4746 (mNeonGreen-ComM, HaloTag-PBP2x), R5297 (mNeonGreen-ComM, RodA-HaloTag), and R5389 (mNeonGreen-ComM, HaloTag-DivIB) were treated with CSP for 10 min before imaging. **a** Bicolor demograph showing the colocalization signals (white) of ComM (green) and PBP2x (magenta) in competent R4746 cells. $N = 4119$ cells. **b** Representative images of R4746 cells in bright field, fluorescence and merged signals obtained by super resolution SIM². mNeonGreen-ComM and HaloTag-PBP2x signals were false colored in green and magenta, respectively. Scale bar, 1 μm. Images representative of two independent biological replicates. **c** Representative SIM² image of a single R4746 cell oriented horizontally. Scale bar, 0.5 μm. Same dataset as in b. **d**–**f** Representative SIM² images of single cells oriented vertically for strains R4746 ($n = 35$ cells) (**d**) R5297 ($n = 45$ cells) (**e**) and R5389 ($n = 89$ cells) (**f**). Data from two independent biological replicates. Scale bars, 0.5 μm. For (**a**–**f**) white color represents colocalized green and magenta signals. **g** Pearson correlation coefficient (PCC) values between fluorescence channels for each protein fusion pair in vertically oriented cells of strains R4746 ($n = 35$), R5297 ($n = 45$) and R5389 ($n = 89$) imaged by SIM. Data are presented as violin plots,

median, 25th and 75th percentiles are shown. Pairwise comparisons were done with a nonparametric two-sided Mann–Whitney test. ****, exact $P$-values < 0.0001. **h**–**j** Ring diameter of PBP2x (**h**), RodA (**i**) and DivIB (**j**) as a function of ComM ring diameter measured in vertically oriented cells imaged by SIM². The red dotted reference line corresponds to X = Y (equal diameter for the two proteins), and the area covered by the red shadow represents the 2.5% confidence interval. The percentage of cells in which the ComM ring is significantly smaller (X < Y) or larger (X > Y) than the PBP2x (**h**), the RodA (**i**) or DivIB (**j**) ring is indicated. Same dataset as in **g**. **k** ComM, PBP2x, DivIB and RodA ring diameters measured in single cells from strains R4746 ($n = 35$), R5389 ($n = 89$) and R5297 ($n = 45$). Same dataset as in g. Values for mNeonGreen-ComM were pooled from the three strains ($n = 169$). Data are presented as violin plots, median and 25th and 75th percentiles are shown. Pairwise comparisons were done with a nonparametric two-sided Mann–Whitney test. ns, $P > 0.05$; **, $0.001 < P < 0.01$. Exact $P$-values were calculated for ComM versus PBP2x ($P = 0.6479$), DivIB ($P = 0.6148$) and RodA ($P = 0.0031$). Source data are provided as a Source data file.

**a**

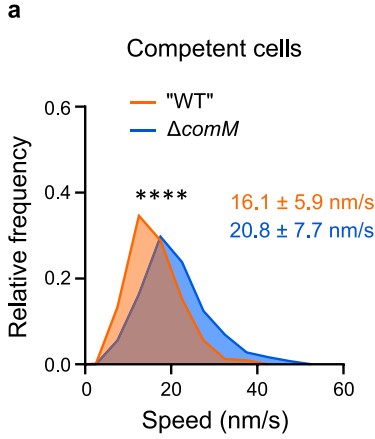

**b**

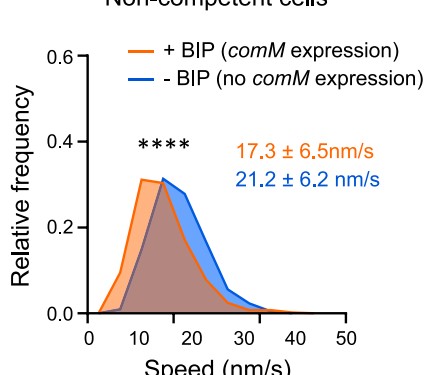

**Fig. 5 | ComM is sufficient to reduce PBP2x speed.** Cells were grown in C + Y medium to early exponential phase and *comM* expression was induced by addition of CSP (competent cells, panel a) or BIP (non-competent cells, panel b) before imaging by time-lapse TIRFM. BIP-1 (noted BIP on the graph for simplicity), is a peptide pheromone able to induce expression of *comM* outside competence in strain R4846 (see Methods section). **a** Distribution of speed of mNeonGreen-PBP2x patches during competence in the Δ*comM* (strain R4845, 538 trajectories) and the wild-type (strain R4743, 644 trajectories) genetic backgrounds. Average speed ± sd

for each strain is indicated. **b** Distribution of speed of mNeonGreen-PBP2x patches in non-competent cells of strain R4846 induced (+ BIP, *comM* expression, 527 trajectories), or not (- BIP, no *comM* expression, 644 trajectories), to produce ComM by BIP induction. Average speed ± sd for each condition is indicated. Data are shown from two independent biological replicates. Pairwise comparisons were done with a nonparametric two-sided Mann–Whitney test. ****, approximate *P*-values < 0.0001. Source data are provided in Source data file.

positive and negative controls, respectively, we used a strain containing LgBit and SmBit fusions to the unrelated DprA protein, which is involved in both homologous recombination[63] and competence shut-off[64] and forms dimers in competent cells[65], and a strain co-expressing the LgBit-ComM fusion with DprA-SmBit. After competence induction, we detected a strong luminescent signal in cells co-expressing the ComM-DivIB and the DprA-DprA pairs, but not in cells co-expressing the ComM-DprA pair (Fig. 6b). Interestingly, we also detected a signal in cells co-expressing LgBit-ComM and FtsW-SmBit that was of similar intensity as the signal in cells co-expressing LgBit-PBP2x and FtsW-SmBit (Fig. 6b), suggesting that ComM and FtsW may interact or are in close proximity in space. However, no signal was detected for the ComM-PBP2x pair (Fig. 6b). Lastly, we performed in vivo non-crosslinked co-immunoprecipitation experiments using strains co-expressing mNeonGreen and ALFA-tag or HaloTag fusions to the proteins of interest. mNeonGreen fusions were immunoprecipitated from detergent-solubilized cell membrane fractions, and co-precipitated proteins were analyzed by immunoblot using anti-ALFA and anti-HaloTag antibodies. PBP2x and FtsW from membrane fractions of either competent or non-competent cultures did not co-immunoprecipitate (Supplementary Fig. 11b), in contrast with a previous report where co-immunoprecipitation was performed under crosslinking conditions[25], suggesting that the FtsW:PBP2x complex is not stable enough to be detected without cross-linking. In this context, the absence of co-immunoprecipitation observed between ComM and FtsW or PBP2x remains inconclusive (Supplementary Fig. 11b). Importantly, however, the ComM-ALFA fusion was repeatedly captured together with mNeonGreen-DivIB, while it was absent in the untagged immunoprecipitation control (Fig. 6c). ALFA-ComM co-immunoprecipitated only partially with mNeonGreen-DivIB (Fig. 6c), suggesting that the complex is not stable or that the two proteins are in different stoichiometry or not always associated in the same complex. In support of the latter, SIM[2] revealed partial colocalisation of mNeonGreen-ComM and HaloTag-DivIB at the division plane (Fig. 4f), confirmed by PCC analysis (Fig. 4g). Finally, measurement of ComM and DivIB ring diameters (Fig. 4j, k) indicated that both proteins are in the same ring at different stages of division. Taken together, these findings suggested that a fraction of ComM and DivIB proteins interact in competent cells.

## Competence induction reduces DivIB speed

DivIB is a bitopic membrane protein that forms a trimeric complex with the divisome proteins FtsL and DivIC (named FtsQBL in Gram-negative bacteria)[66–68]. Notably, evidence suggests that the DivIB/DivIC/FtsL subcomplex activates the septal PG synthases in *Pseudomonas aeruginosa*[47], *Staphylococcus aureus*[46], and *S. pneumoniae*[67]. In *B. subtilis*, DivIB, DivIC, FtsL, FtsW and its cognate bPBP PBP2B were shown to move with similar velocities around the division site[69]. Furthermore, it was recently reported that in *S. aureus* FtsW and DivIB move around the division site with the same speed, which correlates with septum constriction rate[28]. Consistently, we found that a functional mNeonGreen-DivIB fusion (Supplementary Fig. 3, 5) moved with similar velocity than PBP2x and FtsW fusions in exponentially growing pneumococcal cells (Fig. 7a, Supplementary Table 1, and Supplementary Movie 8). In competent cells, DivIB speed decreased to match that of FtsW, PBP2x and ComM (Fig. 7a, b, and Supplementary Table 1). Altogether, our results indicate that the development of competence causes a deceleration of DivIB in a manner similar to that of PBP2x and FtsW.

## Overproduction of DivIB suppresses the division delay of competent cells

Altogether, our results raised the interesting hypothesis that ComM might inhibit DivIB via a direct protein-protein interaction to repress the activity of the septal PG synthase complex FtsW:PBP2x. In this scenario, artificial overproduction of DivIB in competent cells should bypass ComM inhibition and suppress the division delay linked to deceleration of FtsW:PBP2x. When the cellular amounts of DivIB were increased by expressing an additional copy of *divIB* from the IPTG-inducible promoter $P_{Lac}$, no delay in cell division was detected in competent cells relative to non-competent cells (Fig. 7c), similar to what is observed in the Δ*comM* mutant background[38] (Supplementary Fig. 2a). Overproduction of DivIB also restored partially the elongated phenotype caused by the inhibition of septal PG synthesis in competent cells[38,39] (Supplementary Fig. 12a). Furthermore, in competent cells, PBP2x speed was slightly increased in the presence of IPTG, compared to its speed in the absence of inducer (Fig. 7d), while overproduction of DivIB did not affect the speed of PBP2x in non-competent cells (Supplementary Fig. 12b). Altogether, our results

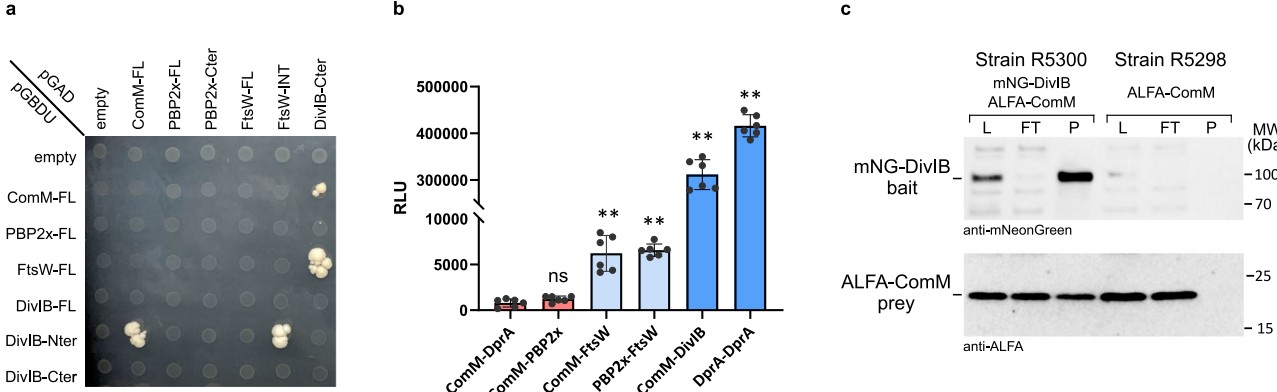

**Fig. 6 | DivIB interacts with ComM, and its overproduction suppresses the competence-dependent cell division delay. a** Yeast-two-hybrid assay to test interactions between ComM, PBP2x, FtsW and DivIB. FL, full length; PBP2x-Cter, C-terminal domain of PBP2x (aa 49–750/750); FtsW-INT, internal domain of FtsW (aa 36–83/409); DivIB-Nter, N-terminal domain of DivIB (aa 1–130/396); DivIB-Cter, C-terminal domain of DivIB (aa 150–396/396). AD, Gal4 activation domain; BD, Gal4 DNA binding domain. The AD and BD domains alone (empty vectors) were used as control for autoactivation. The selective plates lack leucine, uracil and adenine (-LUA). Data are representative of two biological replicates and three technical replicates. **b** Split-luciferase assays on competent cells. Strains R4858, expressing DprA-LgBit and DprA-SmBit, and R5313, expressing LgBit-ComM and DprA-SmBit, were used as positive and negative controls, respectively. Red, strains that did not reconstitute the luciferase (no interaction); light blue, strains that reconstituted the luciferase (positive interaction); dark blue, strains who reconstituted strongly the luciferase (strong interaction). RLU, relative luminescence units. Data are represented as individual data points with bars and error bars indicating averages ± sd, (n = 6, measured over two independent biological replicates and three technical

replicates for each strain). Standard deviations are indicated. Pairwise comparisons were done with strain R5313 as a reference (nonparametric two-sided Mann–Whitney test; ns, $P > 0.05$; **, $0.001 < P < 0.01$). Exact $P$-values were calculated for R5313 versus R5315 ($P = 0.132$), R5312 ($P = 0.0022$), R5317 ($P = 0.0022$), R5285 ($P = 0.0022$) and R4858 ($P = 0.0022$). **c** Immunoblots from co-immunoprecipitation assays of strains R5300 (mNeonGreen-DivIB, ALFA-ComM), and R5298 (ALFA-ComM). Immunoprecipitations using mNeonGreen traps were performed 15 min after competence induction. Detergent solubilized membranes prior to immunoprecipitation (L, load), the supernatants after immunoprecipitation (FT, flow-through), and the immunoprecipitations (IP) were subjected to immunoblot analysis using anti-mNeonGreen (top panels) and anti-ALFA tag antibodies (bottom panels). Equivalent amounts of the load and supernatant (∼3 ml equivalent culture) and approximately five (∼14 ml equivalent culture for anti-mNeongreen immunoblot analysis) and six (∼17 ml equivalent culture for anti-ALFA immunoblot analysis) times more of the IP were analyzed. Representative data of three independent experiments. Source data are provided as a Source data file.

suggest that the ComM-DivIB interaction detected both in the yeast and in pneumococcal cells is physiologically relevant, and point towards a model in which ComM reduces FtsW:PBP2x activity via DivIB. Interestingly, the amino acid sequence of ComM possesses one of the three conserved motifs found in the catalytic site of the so-called CAAX proteases[43] (also called the Abi family proteins[70]). Furthermore, mutations in this motif and in several conserved residues that could be part of a CAAX-like catalytic site abolished the immunity of competent cells toward the PG hydrolase CbpD[43], suggesting that ComM might have a protease activity. We then analyzed the levels of DivIB at different times after competence induction in both wild-type and ΔcomM cells (Supplementary Fig. 13). Western blot analysis indicated that the levels of the mNeonGreen-DivIB fusion remain stable during the course of competence both in the presence and in the absence of ComM. We concluded that ComM interferes with DivIB activity to inhibit cell division in a degradation independent mechanism.

## Discussion

Our findings indicate that the transient inhibition of cell division that occurs during competence in *S. pneumoniae* results from ComM-dependent deceleration of FtsW:PBP2x (Fig. 3a–c), the SEDS:bPBP pair providing PG polymerase and crosslinking activity in the septal PG synthetic machinery. Since PG synthesis activity powers FtsW:PBP2x circumferential motion[25], transient deceleration of FtsW:PBP2x can therefore account for the reduced septal PG synthesis observed in competent cells[38]. In contrast, the speed of the SEDS:bPBP pair RodA:PBP2b, which is reflective of peripheral (sidewall) PG synthesis[53], remains unaffected during competence (Fig. 3a–c). Thus, the elongated phenotype of competent cells[38] may be explained by the unbalance between septal and peripheral PG synthesis. The toxicity associated with uncontrolled overexpression of ComM, reported to result in growth arrest, cell elongation and septum synthesis inhibition -suggested to result from compromised coordination of septal and

lateral cell wall synthesis[43], is also consistent with a role of ComM as FtsW:PBP2x inhibitor. FtsZ treadmilling also remains unaffected during competence and is faster than the speeds of the two PG synthetic machineries (Figs. 2 and 3d), consistent with what was observed in growing (non-competent) *S. pneumoniae* cells[25,53].

Our data also suggests that the competence-specific membrane protein ComM shares the same motor with the FtsW:PBP2x septal PG synthase complex. More importantly, ComM was found to interact with the cell division regulator DivIB using three complementary interactomic approaches: Y2H assays, split-luciferase assays and co-immunoprecipitation (Fig. 6a–c). DivIB, DivIC and FtsL, and their orthologues FtsQ/FtsB/FtsL, form a trimeric transmembrane complex conserved in both Gram-positive and Gram-negative bacteria, respectively[66,71]. Interestingly, the orthologous FtsQBL complex was recently shown to interact with the septal PG synthase and regulate its enzymatic activities[47]. Our finding that DivIB moves around the division site at the same speed as PBP2x and FtsW in both non-competent and competent *S. pneumoniae* cells (Fig. 7a, b) aligns with recent reports showing that DivIB exhibits the same dynamics as the septal PG synthase complex in *S. aureus*[28] and *B. subitlis*[69]. In this context, our findings support a model in which ComM interacts with DivIB to reduce the activity of the FtsW:PBP2x complex, leading to a reduction in septal PG synthesis and, consequently, inhibition of cell division in competent cells (Fig. 8).

It has been previously suggested that ComM may have protease activity[43]. However, the protein levels of FtsW, PBP2x and DivIB were found to be similar in competent and non-competent cells (Supplementary Fig. 1a), and DivIB levels remained stable and comparable in both wild-type and ΔcomM cells during the course of competence (Supplementary Fig. 13). These results indicate that ComM does not affect the levels or processing of the septal PG synthases or DivIB. Although we cannot exclude that ComM degrades an unidentified activator of DivIB, our results argue against a proteolytic role of ComM

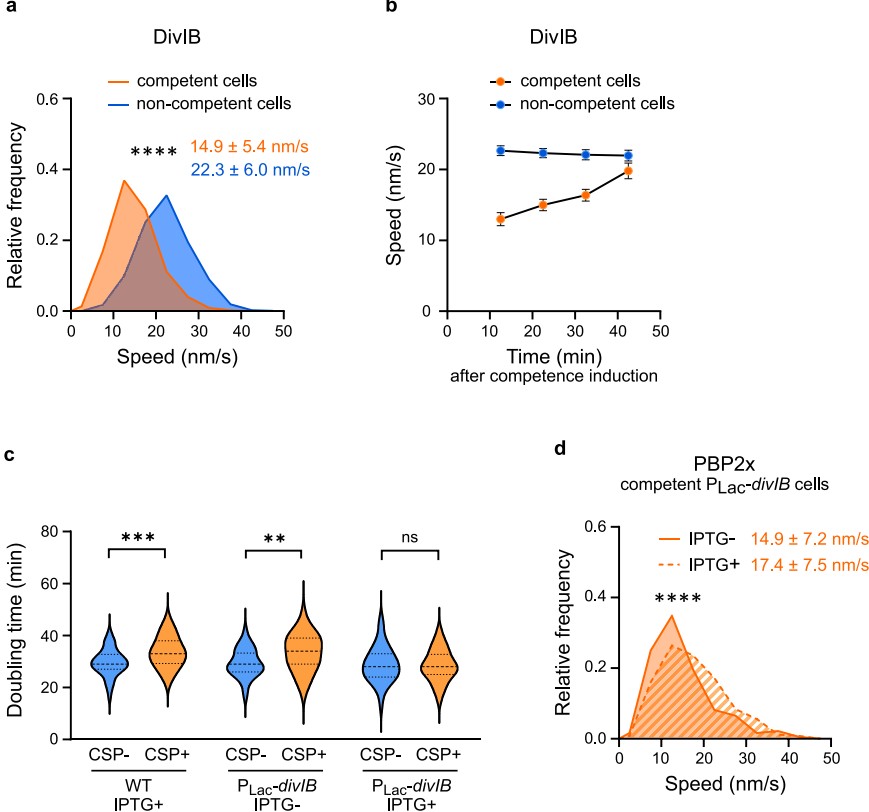

**Fig. 7 | DivIB overproduction suppresses the competence-dependent cell division delay. a** Distribution of speed of DivIB-mNeonGreen (strain R4869) in competent and non-competent cells (450 and 1111 trajectories, respectively, analyzed over two independent biological replicates). Average speed ± sd for each condition is indicated. ****, approximate $P$-value < 0.0001 (nonparametric two-sided Mann–Whitney test). **b** DivIB-mNeonGreen average speed ± sd measured at different time points after CSP addition (competent cells, 627 trajectories) or not (non-competent cells, 1111 trajectories). Error bars represent 95% confidence interval. Same dataset as in a. **c** Doubling time of individual cells from strains R1501 (wild-type) and R5224 ($P_{Lac}$-$divIB$) overexpressing (IPTG +) or not (IPTG-) DivIB. Data are presented as violin plots where the top and bottom dashed lines correspond to the 25th and 75th percentiles, and the middle-dashed line corresponds to the median. The number of cells analyzed with and without CSP, respectively, is as follows: 40 and 112 for R1501 treated with IPTG; 31 and 146 for R5224 in the absence of IPTG; and 92 and 150 for R5224 treated with IPTG. Pairwise comparisons were done with a nonparametric two-sided Mann–Whitney test; ns, $P > 0.05$; **, $0.001 < P < 0.01$; ***, $0.001 < P < 0.0001$. Exact $P$-values were calculated for competent versus non competent cultures of strains R1501 ($P = 0.0002$), R5224 ($P = 0.0014$), and R5224 treated with IPTG ($P = 0.9056$). **d** Distribution of speed of mNeonGreen-PBP2x in competent cells overexpressing (IPTG +, 250 trajectories) or not (IPTG-, 184 trajectories) DivIB (strain R5225). Cells were grown to early exponential phase in C + Y medium supplemented, or not, with IPTG to induce $divIB$ expression, and incubated with CSP for 10 min before imaging. Average speed ± sd measured over two independent biological replicates for each condition is indicated. ****, approximate $P$-value < 0.0001 (nonparametric two-sided Mann–Whitney test). Source data are provided as a Source data file.

in septal PG complex deceleration in competent cells. Alternatively, ComM may have lost its proteolytic activity but retained the ability to bind its targets, potentially modulating the activity of DivIB by direct protein-protein interaction. While the conserved catalytic motif of CAAX proteins was found essential to confer bacteriocin self-immunity in various Gram-positive bacteria[70], other homologs have been proposed to act as a catalytically inactive scaffold for membrane proteins involved in cell envelope synthesis[72,73] or virulence[74]. In particular, in *S aureus*, the transmembrane LyrA protein of the CAAX type II family has been suggested to modulate the activity of the SagB PG hydrolase via the proven interaction through its transmembrane domain[73] and not through a protease activity[72].

While some mechanisms of cell division inhibition affect the assembly or stability of the Z-ring[10,12], most known mechanism involve membrane proteins interacting with FtsW or regulators of the septal synthesis machinery[13–19]. Previous studies have revealed that the mode of action of the DNA damage-induced cell division inhibitors YneA in *B. subtilis*[13], and SidA and DidA in *Caulobacter crescentus*[14,15], involves interactions with either FtsL, FtsW, or with the essential Gram-negative cell division protein FtsN. Similarly, it was shown that the gene product 56 encoded by the lytic bacteriophage SP01 blocks cell division in *B. subtilis* by binding FtsL and interfering with the recruitment of the

septal PG synthase[75]. Midcell recruitment of the FtsW:bPBP septal PG synthase complex and its regulators occur at the latest stages of the divisome assembly, both in Gram-negative and Gram-positive bacteria[76–79]. Targeting and regulating the activity of late-acting proteins of the cell division machinery may be more efficient and cost-effective, in terms of both setup time and energy, than disassembling the entire divisome by targeting the Z-ring. This would be particularly relevant to processes involving a transient inhibition of cell division, such as competence.

Finally, the role of ComM as regulator of the activity of the septal PG synthase complex FtsW:PBP2x is fully compatible with its mode of action as the fratricide immunity protein[80]. During the predatory DNA acquisition process of fratricide, competent cells secrete the muralytic fratricin CbpD, which lyses non-competent sister cells by binding to the septal region[81] and cleaving nascent septal PG[82], while competent cells are protected by ComM by an unknown mechanism. Importantly, evidence suggests that ComM likely mediates immunity indirectly, rather than through direct interaction with CbpD[43]. Based on our findings, a plausible scenario is that by reducing the speed of the septal PG synthase in competent cells, ComM decreases its activity and thus limits the availability of CbpD substrates. However, since mutations in the CAAX-like catalytic site of ComM did abolish immunity to CbpD, a

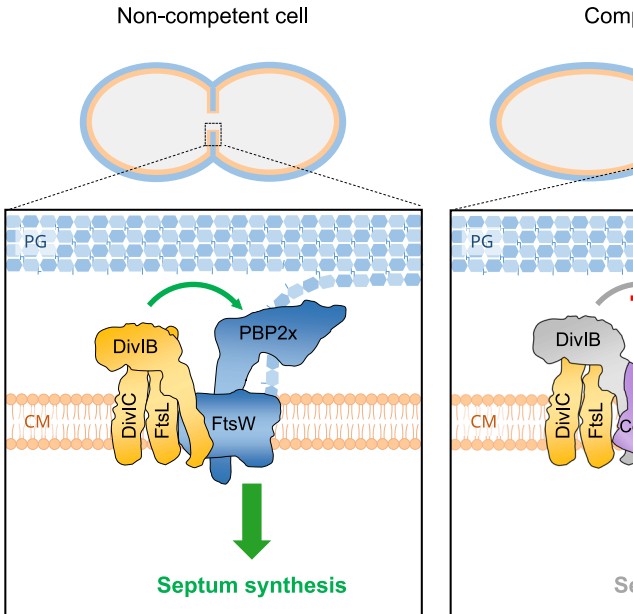

**Fig. 8 | Model for the regulation of septal PG synthesis by ComM via DivIB.** In non competent cells, the DivIB/DivIC/FtsL subcomplex interacts with the septal PG synthase complex FtsW:PBP2x at the site of the division. DivIB activates septal PG synthesis and allows cell division. In competent cells, ComM is expressed and interacts with DivIB to repress its activity, leading to deceleration of septal PG synthesis and the arrest of the cell division. PG: Peptidoglycan, CM: Cell membrane.

catalytic activity of ComM toward protein(s) related to CbpD function, independent of its interaction with DivIB, cannot be ruled out.

Altogether, our findings provide a better mechanistic understanding of the transient inhibition of cell division that halts cell cycle progression in competent pneumococcal cells and raise intriguing questions about the relative regulatory mechanisms of the different types of PG synthetic machineries. Mechanistic insight provided in this study and future work on the modulation of the activities of the elongasome and the divisome will pave the road toward the discovery of novel ways to disrupt these essential cellular processes for antibiotic development.

## Methods
### General methods
*S. pneumoniae* strains and oligonucleotide primers used in this study are listed in Supplementary Tables 1 and 2, respectively. *S. pneumoniae* strains were all constructed in the R1501 background, which is derived from strain R800[83]. This strain contains the ΔcomC mutation that renders it unable to develop competence spontaneously[50]. Synchronicity of competence induction in the entire population is then achieved by addition of synthetic CSP1[35]. Stock cultures were routinely grown at 37 °C to $OD_{550}$ ~ 0.3 in Todd-Hewitt medium (BD Diagnostic System) supplemented with 0.5% Yeast Extract (THY) or in C + Y medium[84]; after addition of 15% (vol/vol) glycerol, stocks were kept frozen at -70 °C. For the monitoring of growth, pre-cultures grown in C + Y medium to $OD_{550}$ ~ 0.3 were diluted 1 in 40 in C + Y medium and further grown to $OD_{550}$ ~ 0.1. Cells were then diluted 40-folds in C + Y medium and distributed into a 96-well microplate (300 μL per well). $OD_{492}$ values were recorded throughout incubation at 37 °C in a Varioskan Flash luminometer (Thermo Fisher ScientificWaltham, MA, USA). Note that we measured OD at 492 nm in the Varioskan luminoter, which gives similar results to monitor cell growth as measurements at OD 550 nm. Transformation experiments[85] were performed using cells grown in C + Y medium at 37 °C to $OD_{550}$ ~ 0.1 and treated for 10 min with CSP1 (100 ng mL⁻¹). Upon addition of transforming DNA, cells were incubated for 20 min at 30 °C. Transformants were plated in 10 mL CAT-agar supplemented with 4% horse blood, incubated for 2 h at 37 °C for phenotypic expression, and selected using a 10 mL CAT-agar overlay containing the appropriate antibiotic. Antibiotic concentrations (μg ml⁻¹) used for the selection of *S. pneumoniae* transformants were: kanamycin (Kan), 250; spectinomycin (Spec), 100; and streptomycin (Str), 200. Ectopic expression of *comM* (strain R4846), *divIB* (strains R5224 and R5225) and *dprA*-SmBiT (strains R4858 and R5313) were achieved by adding 250 μg ml⁻¹ BIP-1[86] and 100 μM IPTG, respectively, to the cultures. BIP-1[86] (Bacteriocin-Inducing Peptide) is a peptide pheromone able to induce expression of genes under the control of the $P_R$ promoter at the inducible expression platform $CEP_R$[87] (see below).

### Strains construction
Strain R4599, containing a *ftsZ-mNeonGreen* fusion at the *ftsZ* endogeneous locus, was generated as follows. Primers were designed to amplify PCR products containing: (I) the upstream region and the entire coding sequence of the *ftsZ* (*spr1510*) gene (2044 bp; oligonucleotides OEC48 and DJ47, and R1501 DNA as template); (II) a 12 amino-acid linker, L5[25,88], and the *orf* of the *mNeonGreen* gene (770 bp; oligonucleotides DJ45 and DJ46, and plasmid DNA pPEPY-$P_{Lac}$-Link-mNeonGreen[89] as template); and (III) the downstream region of the *ftsZ* gene (2048 bp; oligonucleotides DJ48 and OEC49, and R1501 DNA as template). The PCR products were gel-purified and used as templates in a SOEing PCR using the outer primers OEC48 and OEC49. The resulting SOEing PCR product was subsequently used to transform strain R1501 without selection. For this, phenotypic expression was performed in liquid culture for 4 h at 37 °C, and aliquots of the transformed culture were plated without selection. 20 independent colonies were isolated and presence of the gene encoding the fusion protein was confirmed by DNA sequencing.

Strain R4601, harboring a *mNeonGreen* fusion at the *comM* endogenous locus, was created as follows. Primers were designed to amplify PCR products containing: (I) the upstream region of the *comM* (*spr1762*) gene (2049 bp; oligonucleotides DJ14 and DJ13, and R1501 DNA as template); (II) the *orf* of the *mNeonGreen* gene followed by a 6 amino-acid linker, L1[52], yielding a 776 bp PCR fragment (oligonucleotides DJ11 and DJ12, and plasmid DNA pPEPY- $P_{Lac}$-Link-mNeonGreen[89] as template); and (III) the *comM* orf and its downstream region (2154 bp; oligonucleotides DJ15 and DJ16, and R1501 DNA as template).

The PCR products were gel-purified and used as templates in a SOEing PCR using the outer primers DJ14 and DJ16. The resulting SOEing PCR product was subsequently used to transform strain R1501 without selection.

Strain R4728, containing a *ftsW-mNeonGreen* fusion at the *ftsW* endogeneous locus, was generated as follows. Primers were designed to amplify PCR products containing: (I) the upstream region and the entire coding sequence of the *ftsW* (*spr0973*) gene (2043 bp; oligonucleotides OCN440 and OCN441, and R1501 DNA as template); (II) the L5 linker[88] and the *orf* of the *mNeonGreen* gene (804 bp; oligonucleotides OCN442 and OCN443, and plasmid DNA pPEPY-P_{Lac}-Link-mNeonGreen[89] as template); and (III) the downstream region of the *ftsW* gene (2168 bp; oligonucleotides OCN444 and OCN445, and R1501 DNA as template). The PCR products were gel-purified and used as templates in a SOEing PCR using the outer primers OEC440 and OEC445. The resulting SOEing PCR product was subsequently used to transform strain R1501 without selection.

Strain R4743, carrying a *mNeonGreen-pbp2x* fusion, was created as follows. Primers were designed to amplify PCR products containing: (I) the upstream region of the *pbp2x* (*spr0304*) gene (2111 bp; oligonucleotides OCN450 and OCN451, and R1501 DNA as template); (II) the *orf* of the *mNeonGreen* gene and the L1 linker[52] yielding a 776 bp PCR fragment (oligonucleotides OCN452 and OCN453, and plasmid DNA pPEPY-P_{Lac}-Link-mNeonGreen[89] as template); and (III) the *pbp2x* orf and its downstream region (2128 bp; oligonucleotides OCN454 and OCN455, and R1501 DNA as template). The PCR products were gel-purified and used as templates in a SOEing PCR using the outer primers OCN450 and OCN455. The resulting SOEing PCR product was subsequently used to transform strain R1501 without selection.

Strain R4744, containing a *HaloTag-pbp2x* fusion, was created as follows. Primers were designed to amplify PCR products containing: (I) the upstream region of the *pbp2x* gene (2100 bp; oligonucleotides OCN450 and OCN456, and R1501 DNA as template); (II) a 963 bp PCR fragment carrying the *orf* encoding the HaloTag sequence and a 15 amino-acids linker, L6[25] (oligonucleotides OCN457 and OCN458, and a DNA fragment containing the HaloTag sequence generated by Integrated DNA Technologies, see below); and (III) the *pbp2x* orf and its downstream region (2127 bp; oligonucleotides OCN459 and OCN455, and R1501 DNA as template). The PCR products were gel-purified and used as templates in a SOEing PCR using the outer primers OCN450 and OCN455. The resulting SOEing PCR product was subsequently used to transform strain R1501 without selection.

Strain R4746 (Δ*comC*, *mNeonGreen-comM*, *HaloTag-pbp2x*) results from the transformation without selection of strain R4601 with a PCR fragment carrying the *HaloTag-pbp2x* construct (oligonucleotides OCN450 and OCN455, and R4744 DNA as template).

Strain R4845 (Δ*comC*, *mNeonGreen-pbp2x*, *comM::aad9*) results from the transformation of strain R4743 with genomic DNA from strain R1879[38] and selection for transformants in the presence of spectinomycin.

Strain R4846 results from the transformation of strain R4743 with genomic DNA from strain R3957[38] and selection for transformants in the presence of kanamycin. CEP_R is a BIP-1 inducible expression platform described previously[87]. The CEP_R-*comM* (kan) construct[38] allows expression of *comM* outside competence under the control of the P_R promoter inducible by the BIP-1 peptide[86], at the CEP (Chromosomal Expression Platform) locus[90].

Strain R4867 harboring a *rodA-mNeonGreen* fusion was generated as follows. Primers were designed to amplify PCR products containing: (I) the upstream region and the coding sequence of the *rodA* (*spr0712*) gene (2163 bp; oligonucleotides DJ59 and DJ60, and R1501 DNA as template); (II) the L5 linker[88] and the *orf* of the *mNeonGreen* gene (798 bp; oligonucleotides DJ61 and DJ62, and plasmid DNA pPEPY-P_{Lac}-Link-mNeonGreen[89] as template); and (III) the downstream region of the *rodA* gene (2128 bp; oligonucleotides DJ63 and DJ64, and R1501

DNA as template). The PCR products were gel-purified and used as templates in a SOEing PCR using the outer primers DJ59 and DJ64. The resulting SOEing PCR product was subsequently used to transform strain R1501 without selection.

Strain R4869, carrying a *mNeonGreen-divIB* fusion, was created as follows. Primers were designed to amplify PCR products containing: (I) the upstream region of the *divIB* (*spr0605*) gene (2080 bp; oligonucleotides DJ79 and DJ80, and R1501 DNA as template); (II) the *orf* of the *mNeonGreen* gene followed by the 6 amino-acid linker L1[52], yielding a 775 bp PCR fragment (oligonucleotides DJ81 and DJ82, and plasmid DNA pPEPY-P_{Lac}-Link-mNeonGreen[89] as template); and (III) the *divIB* orf and its downstream region (2510 bp; oligonucleotides DJ83 and DJ84, and R1501 DNA as template). The PCR products were gel-purified and used as templates in a SOEing PCR using the outer primers DJ79 and DJ84. The resulting SOEing PCR product was subsequently used to transform strain R1501 without selection.

To over-produce DivIB, we constructed a strain, R5224 (P_{Lac}-*divIB*), allowing ectopic expression of *divIB* (*spr0605*) at the CEP chromosomal expression platform[90] under the control of the IPTG-inducible P_{Lac} promoter[91]. Primers were designed to amplify PCR products containing: (I) the 5′ region of the CEP platform including the *lacI* gene and the P_{Lac} promoter (3783 bp; oligonucleotides OCN564 and OCN560, and R3310[91] DNA as template); (II) the *orf* of the *divIB* gene (1192 bp; oligonucleotides OCN561 and OCN562, and R1501 DNA as template); and (III) the 3′ region of the CEP platform including the gene conferring resistance to kanamycine (2899 bp; oligonucleotides OCN563 and OEC111, and R3310 DNA as template). The PCR products were gel-purified and used as templates in a SOEing PCR using the outer primers OCN564 and OEC111. The resulting SOEing PCR product was subsequently used to transform strain R1501 and clones resisting to kanamycin were selected.

Strains R5298, R5300 and R5385, harboring an *ALFA-comM* fusion at the *comM* endogenous locus, were created as follows. Primers were designed to amplify PCR products containing: (I) the upstream region of the *comM* (*spr1762*) gene and 27 nucleotides at the 5′ extremity of the 39 bp ALFA-Tag sequence[92] (2051 bp; oligonucleotides DJ14 and OCN647, and R1501 DNA as template); and (II) a 2168 bp PCR fragment carrying 36 nucleotides at the 3′ extremity of the ALFA-Tag sequence followed by the L1 linker[52], the *comM* orf and its downstream region (oligonucleotides OCN609 and DJ16, and R1501 DNA as template). Note that annealing of oligonucleotides OCN647 and OCN609 reconstitutes the full sequence encoding the ALFA-Tag flanked by methionine and proline codons at the 5′ and 3′ extremities respectively, as described[92]. The PCR products were gel-purified and used as templates in a SOEing PCR using the outer primers DJ14 and DJ16. The resulting SOEing PCR product was subsequently used to transform without selection strains R1501 ("WT"), R4869 (carrying a *mNeonGreen-DivIB* fusion), and R4728 (containing a *ftsW-mNeonGreen* fusion), yielding strains R5298, R5300 and R5385, respectively.

Strain R5386 (Δ*comC*, *ftsW-mNeonGreen*, *HaloTag-pbp2x*) results from the transformation without selection of strain R4728 with a PCR fragment carrying the *HaloTag-pbp2x* construct (oligonucleotides OCN450 and OCN455, and R4744 DNA as template).

Strain R5389, harboring a Halo-Tag fusion at the *divIB* endogeneous locus, was created as follows. Primers were designed to amplify PCR products containing: (I) the upstream region of the *divIB* (*spr0605*) gene (2080 bp; oligonucleotides DJ79 and OCN632, and R1501 DNA as template); (II) a 963 bp PCR fragment carrying the *orf* encoding the HaloTag sequence and a 15 amino-acids linker, L6[25] (oligonucleotides OCN633 and OCN634, and a DNA fragment containing the HaloTag sequence generated by Integrated DNA Technologies, see below); and (III) the *divIB* orf and its downstream region (2510 bp; oligonucleotides OCN635 and DJ84, and R1501 DNA as template). The PCR products were gel-purified and used as templates in a SOEing PCR using the outer primers DJ79 and DJ84. The resulting SOEing PCR

product was subsequently used to transform strain R4601 without selection.

The R5276, R5285, R5292, R5311, R5312, R5315 and R5317 following strains were used to perform the Split-luciferase complementation assay, which relies on the reconstitution of the bioluminescence luciferase Nanoluc enzyme when a pair of proteins fused to a large Nanoluc bit (LgBit) and a small complementary bit (SmBit) interact or are in close proximity in cells (see below)[62].

Strain R5276, harboring a *LgBiT-comM* fusion at the *comM* endogenous locus, was created as follows. Primers were designed to amplify PCR products containing: (I) the upstream region of the *comM* (*spr1762*) gene (2049 bp; oligonucleotides DJ14 and oIM282, and R1501 DNA as template); (II) the *orf* of the *LgBiT* gene followed by linker L1[52], yielding a 543 bp PCR fragment (oligonucleotides oIM283 and oIM284, and R4858[93] DNA as template); and (III) the *comM orf* and its downstream region (2154 bp; oligonucleotides oIM285 and DJ16, and R1501 DNA as template). PCR products were purified and used as templates in a SOEing PCR using the outer primers DJ14 and DJ16. The resulting SOEing PCR product was gel purified and subsequently used to transform strain R1501 without selection.

Strain R5285, harboring a *LgBiT-comM* fusion at the *comM* endogenous locus and a *SmBiT-divIB* fusion at the *divIB* endogenous locus, was created as follows. Primers were designed to amplify PCR products containing: (I) the upstream region of the *divIB* (spr0605) gene (2080 bp; oligonucleotides DJ79 and oIM286, and R1501 DNA as template); (II) the SmBiT domain followed by the L1 linker[52] and the *divIB orf* and its downstream region (2539pb; oligonucleotides oIM245 - this primer includes the linker-*SmBiT* sequence—and DJ84, and R4869 DNA as template). The two PCR products were purified and used as templates in a SOEing PCR using the outer primers DJ79 and DJ84. The resulting SOEing PCR product was gel purified and subsequently used to transform strain R5276 without selection.

Strain R5292, harboring a *LgBiT-comM* fusion at the *comM* endogeneous locus and CEP_{Lac}-*dprA-SmBit* at the *ami* locus was created as follows. The CEP_{Lac}-*dprA-SmBit* region was amplified from strain R4858 using primer pairs oEC88-oIM149. The PCR product was purified and used to transform strain R5276 and transformants were selected with kanamycin.

Strain R5311, harboring a *ftsW-SmBiT* fusion at the *ftsW* endogeneous locus, was created as follows. Primers were designed to amplify PCR products containing: (I) the upstream region and the entire coding sequence of the ftsW (spr0973) gene and the L5 linker to the *SmBiT* domain (2062 bp; oligonucleotides oCN440 and oIM290 - this primer includes the linker-SmBiT sequence - and R4728 DNA as template); (II) the downstream region of *ftsW* (2143pb; oligonucleotides oIM291 - this primer includes the linker-SmBiT sequence - and oCN445, and R1501 DNA as template). The PCR products were purified and used as templates in a SOEing PCR using the outer primers oCN440 and oCN445. The resulting SOEing PCR product was gel purified and subsequently used to transform strain R1501 without selection.

Strain R5312, harboring a *LgBiT-comM* fusion at the *comM* endogeneous locus and a *ftsW-SmBiT* fusion at the *ftsW* endogeneous locus, was created as follows. Primers were designed to amplify PCR products containing: (I) the upstream region and the entire coding sequence of the ftsW (spr0973) gene and the L5 linker to the *SmBiT* domain (2062 bp; oligonucleotides oCN440 and oIM290 - this primer includes the linker-SmBiT sequence - and R4728 DNA as template); (II) the downstream region of *ftsW* (2143pb; oligonucleotides oIM291 - this primer includes the linker-SmBiT sequence - and oCN445, and R1501 DNA as template). The two PCR products were purified and used as templates in a SOEing PCR using the outer primers oCN440 and oCN445. The resulting SOEing PCR product was gel purified and subsequently used to transform strain R5276 without selection.

Strain R5315, harboring a *LgBiT-comM* fusion at the *comM* endogeneous locus and a *SmBiT-pbp2x* fusion at the *pbp2x* endogeneous locus, was created as follows. Primers were designed to amplify PCR products containing: (I) the upstream region of the *pbp2x* (*spr0304*) gene (2108 bp; oligonucleotides oCN450 and oIM292, and R1501 DNA as template); (II) the *SmBiT* domain followed by a 6 amino-acid linker, LEGSG, and a large part of the *pbp2x* orf (2157pb; oligonucleotides oIM245 - this primer includes the linker-SmBiT sequence - and oCN455, and R4743 DNA as template). The PCR products were purified and used as templates in a SOEing PCR using the outer primers oCN450 and oCN455. The resulting SOEing PCR product was gel purified and subsequently used to transform strain R5276 without selection.

Strain R5317, harboring a *ftsW-SmBiT* fusion at the *ftsW* endogeneous locus and a *LgBiT-pbp2x* fusion at the *pbp2x* endogeneous locus, was created as follows. Primers were designed to amplify PCR products containing: (I) the upstream region of the pbp2x (spr0304) gene (2108 bp; oligonucleotides oCN450 and oIM293 and R1501 DNA as template); (II) the *orf* of the *LgBiT* gene followed by a 6 amino-acid linker, LEGSG, yielding a 543 bp PCR fragment (oligonucleotides oIM294 and oIM295, and R4858 DNA as template); (III) a large part of the *pbp2x* orf (2106 pb; oligonucleotides oIM296 and oCN455, and R4743 DNA as template). PCR products were purified and used as templates in a SOEing PCR using the outer primers oCN450 and oCN455. The resulting SOEing PCR product was gel purified and subsequently used to transform strain R5311 without selection.

All constructs generated by PCR were confirmed by DNA sequencing.

The DNA fragment containing the HaloTag sequence was purchased from Integrated DNA Technologies (IDT). This fragment carries the HaloTag sequence codon optimized for *S. pneumoniae* published with the Winkler laboratory[25] with a few modifications. It also contains the L5 and L6 linkers at the 5' and 3' extremities, respectively (Supplementary Fig. 14).

## Whole-cell extracts preparation and immunoblot analysis

Pneumococcal cells were grown in C + Y medium at 37 °C to exponential phase (OD_{550} ∼ 0.1) and treated or not with CSP for 15 min, and samples (3 ml) were collected by centrifugation. Cell pellets were stored at -80 °C. Whole-cell extracts were prepared by resuspension of cell pellets in 50 µl lysis buffer [10 mM Tris pH 8.0, 1 mM EDTA, 0.01% (wt/vol) DOC, 0.02% (wt/vol) SDS] and incubation at 37 °C for 10 min followed by addition of 12.5 µl 5X loading buffer [312.5 mM Tris pH 6.8, 10% (wt/vol) SDS, 50% (vol/vol) Glycerol, 750 mM β-mercaptoethanol, 0.05% (wt/vol) Bromophenol bue]. Samples were heated for 15 min at 50 °C prior to loading.

For immunoblot analysis, we loaded 10 µl of whole cell extracts resulting from cultures treated with CSP (corresponding to ∼480 µl of cells) and equivalent volumes of non-treated cells normalized by OD550. Samples were loaded on Biorad mini-PROTEAN TGX stain-free 4–15% pre-cast gels. Gels were activated by ultraviolet (UV) exposure for 45 s using a Bio-Rad ChemiDoc MP imager to visualize and estimate total protein per lane. Resulting images were further used as loading controls. Proteins were then transferred to a nitrocellulose membrane, using a Turbo Blot transfer unit (Bio-Rad). mNeonGreen, and HaloTag and anti ALFA-Tag fusion proteins were detected using polyclonal anti-mNeonGreen (NC, unpublished), and anti-HaloTag (G921A, Promega) and anti-ALFA (N1581, NanoTag Biotechnologies) antibodies diluted at 1:10,000, 1:5,000 and 1:10,000 respectively. Primary antibodies were detected using peroxidase-conjugated goat anti-rabbit immunoglobulin G (Sigma) diluted 1:10,000. Membranes were further incubated with clarity chemiluminescence substrate (Bio-Rad), and imaged on the ChemiDoc MP. Apart from the mNeonGreen-ComM fusion, detection and measurement of band intensities performed using Image Lab 5.2 software (Bio-Rad), and normalized with respect to total protein quantification[94], indicated that the level of fusion proteins remained similar in non-competent and competent samples.

## Sample preparation for microscopy

After gentle thawing of stock cultures, aliquots were inoculated at $OD_{550}$ 0.006 in C + Y medium and grown at 37 °C to an $OD_{550}$ of 0.3. These precultures were inoculated (1/50) in C + Y medium and incubated at 37 °C to an $OD_{550}$ of 0.1. Then, competence was induced with synthetic CSP1 (50 ng ml$^{-1}$). Ectopic *comM* expression was induced by 250 µg ml$^{-1}$ BIP-1[95,96]. To visualize Halo-Tag fusions, Janelia Fluor® 549 HaloTag® ligand (GA1110, Promega) was added to cells at a final concentration of 20 nM, and incubated for 10 min with no washing step before imaging. 0.8 µl of this suspension was immobilized on a 1.2% C + Y agarose-coated microscope slide. For time-lapse microscopy experiments, cells grown at 37 °C to an $OD_{550}$ of 0.1, were induced with CSP1 (50 ng ml$^{-1}$) for 5 min and 0.8 µl was immediately spotted on a microscope slide containing a slab of 1.2% C + Y agarose and covered with a cover glass (# 1.5) before imaging.

## Vertical cell mounting

For vertical imaging, silicon molds containing pyramid-shaped pillars (1 µm in diameter at the top, 2 µm at the base and 5 µm high) were prepared for inverse replication (technological support of LAAS-CNRS micro and nanotechnology platform, Toulouse, France). Agarose pads were prepared by pouring melted 6% agarose in C + Y medium on slides, subsequently covered with the silica molds to generate microholes. 6 µl of fresh cells were deposited on agarose pads and the slides were centrifuged in a mini-centrifuge (10 s, 2000 *g*). 5 µl of 1% agarose in C + Y medium were then poured on top of the agarose pad to immobilize cells in the microholes, and covered with a cover glass before imaging.

## Total Internal Reflection Fluorescence (TIRF) and Highly Inclined and Laminated Optical sheet (HILO) microscopy

All TIRF and HILO imaging was performed on a Zeiss Elyra PS1 automated inverted microscope with an incubation chamber (temperature set to 37 °C) and equipped with a 100 × /1.46 NA Apochromat oil immersion objective. The excitation of fluorescence was achieved using the 488 nm and 561 nm laser lines at a power output ranging from 10% to 15% of the maximum output power (100 mW). The collected emission signals were subsequently filtered using a bandpass filter set (emission filters at 500 - 575 nm and 576 - 650 nm for the 488 nm and 561 nm laser lines, respectively). Images were captured using an Andor iXon 897 EM-CCD camera with a final pixel size of 64 nm (an additional optovar lens 2.5x was placed before the camera). The camera and microscope were controlled by the ZEN software (ZEN 2012 SP2). For time-lapse experiments, images were acquired at intervals of 3 s or 5 s with an exposure time of 100–150 ms (depending on the strain) and a camera EM gain value of 300x. The Definite Focus was activated in order to compensate for the focus drift of the sample during data acquisitions.

## Epifluorescence microscopy

Epifluorescence images were captured using a Nikon Ti-E automated inverted microscope, equipped with a 100 x /1.3 NA CFI Plan Fluor oil immersion objective and an incubation chamber (temperature set to 37 °C). Excitation of fluorescence was achieved through the use of the 488 nm and 561 nm laser lines, with a power output of 30% of the maximum excitation power (150 and 50 mW, respectively). Subsequently, the collected emission signals were filtered using a band-pass filter set (emission filters at 520/35 nm and 593/40 nm for the 488 nm and 561 nm laser lines, respectively). The images were captured using an Orca R2 Hamamatsu CCD camera, resulting in a final image pixel size of 64.5 nm. The camera and microscope were controlled by the MetaMorph software v7.10.5.476 (Molecular Devices). Depending on the specific strain in question, the exposure time was 500–750 ms.

## Speed measurements of circumferentially moving proteins

Time-lapses image sequences were stabilized using the Fiji plugin "Image stabilizer" in translation mode with a maximum pyramid level of 1, a template update coefficient of 0.90, 200 maximal iterations and an error tolerance of 0.0000001. Speeds of directionally moving proteins were quantified in the stabilized time-lapse series using a script implemented in MATLAB (Mathworks, R2018b). This method automatically generates kymographs distributed at regular intervals (~1 pixel) perpendicular to a manually defined axis (segment connecting the two opposite poles of the bacteria). The kymographs obtained are aligned side-by-side to generate a single 2D image (the vertical axis corresponding to time and the horizontal axis to spatial displacement). Speed is quantified by measuring the slope of the line. For competent cells speed distributions, speed was measured from cells imaged during the first 30 min after competence induction.

For speed measurements of mNeonGreen-ComM on vertically immobilized cells, each image of the stack was denoised by subtracting a minimal projection filtered by a median filter (radius = 2 pixels). Circular kymographs were generated in Fiji (v2.14.0)[97] using the 'Multi Kymograph' plugin. A circle was manually drawn on ComM rings (centered and of the same diameter). The selection was then transformed into a polygonal line to generate kymographs with a 3-pixel linewidth. Finally, velocities were quantified by measuring the slopes on the kymographs.

## Demograph analysis

Demographs showing protein fluorescence intensity as a function of cell length were generated from epifluorescence images by using Microbe J (version 5.13n)[98] with the following parameters: (area [µm$^2$] 0.6-20; length [µm] 0.5-max; width ([µm] 0.5–2; circularity [0–1] 0-max; curvature [0-max] 0-max; sinuosity [0-max] 0-max; angularity [rad] 0-0.25; solidity [0-max] 0.85-max; intensity [0-max] 0-max)). Cells with regular shapes and sizes but excluded from analyses due to close proximity to other cells were manually added back to analyses. In addition, late-divisional cells were verified to correctly count them as two separated cells if necessary[38]. The demographs were normalized as proposed in microbeJ.

## Heat map analysis

Heat map showing protein fluorescence average intensity in the cell were generated by using Microbe J (version 5.13n)[98] with the following parameters as criteria to define the categories of cells: A: [SHAPE.length/SHAPE.width]≤1.5; B: [SHAPE.length/SHAPE.width]>1.5 and [SHAPE.length/SHAPE.width]≤1.7; C: [SHAPE.length/SHAPE.width]>1.7 and [SHAPE.length]≤1.6; D: [SHAPE.length/SHAPE.width]>1.7 and [SHAPE.length]>1.6 E: [SHAPE.length/SHAPE.width]>2 and [SHAPE.circularity]<0.8; F: [SHAPE.length/SHAPE.width]>2 and [SHAPE.circularity]<0.7.

## Lattice SIM imaging and SIM² image reconstruction

Lattice SIM imaging was conducted mainly as previously described[99] using an Elyra 7 AxioObserver (Zeiss) inverted microscope. Here, we used a 63×/NA 1.46 objective (Zeiss, alpha Plan-Apochromat 63x/1.46 Oil Korr M27) yielding a final pixel size of 64.5 nm for raw images. Fluorescence was excited using a 488 nm (100 mW) and a 561 nm (100 mW) laser lines at 80% and 40% of maximal output power for 488 and 561 laser respectively. Exposure time per phase was 100 ms and 200 ms for 488 nm and 561 nm laser respectively. Temperature was maintained at 30 °C during acquisitions.

Lattice SIM image reconstruction was performed via Zen 3.0 SR Software (Zeiss, black edition) using the nonlinear iterative reconstruction algorithm SIM² with 2D+ processing active or the SIM algorithm with 2D processing active. For SIM² reconstruction, general settings were set to "Live" and "Strong" with the following specifics: 20 iterations, a regularization weight of 0.02, a processing sampling of 4×,

and an output sampling of 4×. Advanced filter parameters were set to "Best fit" and "Median" with a sectioning value of 100 and Baseline "Yes". Following SIM² reconstruction, all images had a lateral pixel size of 16.1 nm. For SIM reconstruction, the following adjusted parameters were applied: a sharpness of 6, advanced filters parameters were set to "Best fit" with a sectioning value of 100, advanced settings were set to Baseline Cut "No" and Scale to Raw "Yes". Following SIM reconstruction, all images had a lateral pixel size of 32.2 nm. PCC was calculated with the Fiji plugin "JaCoP" v2.1.121 using the "Pearson's Coefficient" mode on individual rings in SIM images instead of SIM² to avoid truncated intensities and to preserve the linearity of the signal altered by the reconstruction in SIM². PCC = 1 would indicate perfect colocalization; PCC = 0, no colocalization; PCC > 0.5, significant colocalization.

### Single cell doubling time and morphologic quantification

Phase contrast time-lapses image sequences were stabilized using the Fiji plugin "Image stabilizer" as described above for velocity measurements. Segmentation of phase-contrast images was performed using the image segmentation tool Omnipose[100]. Cell morphology (cell width and length) was estimated as the width and length of the minimum area rectangle around the segmented mask. Doubling times were measured from tracks of individual cells generated with the overlap tracker in Trackmate[101]. Tracking graphs were extracted and analyzed using a custom Python script that worked as follows: a division was detected as a node (representing a cell) having 2 children nodes instead of one. A small number of artefacts, primarily resulting from the misdetection of dead elements in the background as cells, were excluded from analysis by removing graph branches with fewer than 5 nodes. Cell division time was measured as the interval between two successive division events. Due to the short acquisition time (1 h) and to avoid biasing results toward shorter division times, we only considered divisions in cells that completed their first division within the first 20 min. Since competence is a transient state lasting approximately 20–30 min, our unbiased automated analysis likely included some cells that were already exiting competence, resulting in a slightly faster doubling time for competent cells than previously reported from manual analysis[38,39].

### Yeast two hybrid (Y2H) assays

To screen a genomic library for Y2H interactions with ComM, the DNA sequence encoding for ComM was PCR-amplified using R1501 DNA as template and primers OCN334 and OCN430, and cloned as in-frame fusion of the Gal4 binding domain (BD) into plasmid pGBDU-C1[102], using the In-Fusion HD cloning kit (Clontech). The resulting pGBDU-comM plasmid was transformed into the *Saccharomyces cerevisiae* strain PJ69-4a to generate the bait strain. Yeast cells expressing BD-ComM were used to screen a genomic *S. pneumoniae* Library fused to the Gal4 activation domain (AD) in plasmid pGAD-C1 in strain PJ69-4α[103]. After mating on rich medium, diploid yeast strains were selected for interaction between the BD-ComM bait and prey proteome fragments on synthetic complete medium[104] lacking leucine, uracil, and either histidine (to select for expression of the HIS3 interaction reporter) or adenine to select for expression of the ADE2 interaction reporter, after 7–14 days[63]. Prey candidates were identified by PCR amplification and sequencing of the pGAD-C1 insert using primers mAD1ext and mBD2ext.

For Y2H matrices, the coding sequences of the selected proteins: FtsW, PBP2x, C-terminal domain of PBP2x (PBP2x-Cter; aa 49-750/750), DivIB, N-terminal domain of DivIB (DivIB-Nter; aa 1–130/396) and the C-terminal domain of DivIB (DivIB-Cter; aa 150–396/396) were PCR-amplified using R1501 DNA as template, and inserted into plasmids pGAD-C1 and pGBDU-C1[102], as in-frame fusions of the Gal4 AD and BD, respectively, using the In-Fusion HD cloning kit (Clontech). Primers used were OCN364 and OCN365 (FtsW) OCN366 and OCN367 (PBP2x),

OCN368 and OCN367 (PBP2x-Cter), OCN346 and OCN347 (DivIB), OCN346 and OCN348 (DivIB-Nter), OCN349 and OCN347 (DivIB-Cter), and OCN423 and OCN424 (PGAD-C1 and PGBDU-C1). The resulting prey and bait plasmids were transformed into the yeast strains PJ69-4α and PJ69-4a, respectively. A *S. cerevisiae* PJ69-4α strain expressing the internal domain of *S. pneumoniae* FtsW ('FtsW-INT (aa 36–83/409; R.C.-L lab collection)) fused to the AD was also used. Diploid *S. cerevisiae* cells were generated by mating on rich medium. Binary interactions were revealed by growth of diploid cells on selective media as above[104]. Controls with empty vector plasmids (*i.e.*, expressing the BD or AD domains alone) were systematically included to test for auto-activation.

### Split-luciferase complementation assay

Pneumococcal cells were grown in C + Y medium (with 50 μM IPTG where required) at 37 °C to an OD$_{550}$ of 0.1, and competence was induced by addition of synthetic CSP1 (50 ng ml$^{-1}$). Cells were further incubated for 15 min at 37 °C before washing in fresh C + Y medium. NanoGlo substrate (Promega) was then added and luminescence was measured 20 times every 30 sec in a plate reader (Varioskan Flash luminometer, Thermo Fisher ScientificWaltham, MA, USA) using the SkanIt software 7.0.2 (Thermo Scientific) for data acquisition. Data are represented as mean ± SEM calculated from three biological replicates with two technical replicates for each, with individual data points plotted.

### Co-immunoprecipitation from Detergent-solubilized Membranes

100-ml cultures were grown in C + Y medium to OD$_{550}$ ~ 0.1, induced to develop competence by addition of synthetic CSP1 (50 ng ml$^{-1}$) and further incubated for 15 min. After centrifugation, cell pellets were resuspended in 10 ml Buffer A (1 M Sucrose, 100 mM Tris-HCl pH8, 2 mM MgCl2, 3 mM β-mercaptoethanol) and incubated for 30 min at 37 °C to produce protoplasts. Protoplasts were collected by centrifugation (3000 g, 30 min, room temperature) and resuspended with 3 ml hypotonic buffer (Buffer B) (150 mM NaCl, 10 mM Tris-HCl pH8, 1 mM dithiothreitol, 1 mM MgCl2, with protease inhibitors (cOmplete™ Roche)). DNAse I (25 μg ml$^{-1}$, Sigma) and RNAse (20 μg ml$^{-1}$, Sigma) were added to lysates and incubated for 1 h on ice. The membrane fraction was separated by centrifugation at 50,000 g for 30 min at 4 °C. The supernatant was carefully removed, and the membrane pellet was dispersed in 200 μl Buffer B. Crude membranes may be conserved at -80 °C for a few weeks. Membranes were then solubilized by the addition of 20 μl of Buffer B containing the nonionic detergent Igepal CA-630 (Sigma) to a final concentration of 0.6%. The mixture was rotated at 4 °C for 1 h. Soluble and insoluble fractions were separated by centrifugation at 16,000 g for 30 min at 4 °C.

40 μl of the soluble fraction from the Igepal CA-630 treated membrane preparation (the Load) were added to 10 μl of 5X sodium dodecyl sulfate (SDS) sample buffer (62.5 mM Tris-HCl pH 6.8, 2% SDS, 10% glycerol, 150 mM β-mercaptoethanol), and the rest was mixed with mNeonGreen-Trap Magnetic Agarose beads (Chromotek) and rotated for 6 h at 4 °C. The beads were washed three times with 500 μl Buffer B containing 0.6% Igepal CA-630 and immunoprecipitated proteins (IP) were eluted by addition of 50 μl 5X SDS sample buffer and heated for 15 min at 50 °C. The load (9 μl, corresponding to ~3 ml of culture), flow-through (9 μl, corresponding to ~3 ml of culture), and immunoprecipitate (9 μl, corresponding to ~14 ml of culture for anti-mNeonGreen immunoblot analysis, and 10.5 μl, corresponding to ~17 ml equivalent culture for anti-ALFA immunoblot analysis) were then analyzed by immunoblot.

### Statistical analyses

All statistical analyses were performed with GraphPad Prism 10 (GraphPad Software, LLC). Pairwise comparison between two

conditions were done with a two-sided nonparametric Mann–Whitney test, *P*-values are displayed as follows: ****, $P < 0.0001$; ***, $0.0001 < P < 0.001$; **, $0.001 < P < 0.01$; *, $0.01 < P < 0.05$; ns, $P > 0.05$. Note that GraphPad Prism computes an exact *P*-value when the size of the smallest sample is less than or equal to 100, and approximates the *P*-value from a Gaussian approximation when the samples are large.

## Reporting summary

Further information on research design is available in the Nature Portfolio Reporting Summary linked to this article.

## Data availability

Source data are provided with this paper. Research data (fluorescent images and all movies used to measure speeds of single particles including 10–15 examples of kymographs per condition) have been deposited in the public repository platform Zenodo (https://doi.org/10.5281/zenodo.15394703)[105]. All relevant data are available in this article and its Supplementary information files. Source data are provided with this paper.

## Code availability

The analysis code used in this study for single cell doubling time and morphologic quantification was written in Python (v3.10) and is deposited on GitHub at https://github.com/aurelien-barbotin/proced-deepseg[106]. The analysis codes used in this study for speed measurements of circumferentially moving proteins were written in MATLAB language for horizontally oriented cells and in Fiji/ImageJ scripts for vertically oriented cells and are deposited on GitHub at https://github.com/CyrilleBillaudeau/Kymo_Analyser_MultiChannel[107] and https://github.com/CyrilleBillaudeau/CircleKymoIJ[108], respectively.

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

## Acknowledgements

This work is part of DJ's PhD thesis, co-supervised by N.C and R.C.-L. We thank Jan-Willem Veening for plasmid pPEPY-P_Lac-Link-mNeonGreen[89]

and Christophe Grangeasse for strain WT-gfp-pbp2b[52]. We also thank Anne-Lise Soulet, Léa Wagner and Maria-Victoria Prejean for their experimental assistance. Silicon molds containing pyramid-shaped pillars for vertical imaging were produced by the LAAS-CNRS micro and nanotechnologies platform member of the French RENATECH network. This work was funded by grants from the Agence Nationale de la Recherche (EXStasis, ANR-17-CE13-0031 to P.P., N.C. and R.C.-L.) and the European Research Council (ERC) under the Horizon 2020 research and innovation program (grant agreement No 772178 to R.C.-L.), by the Centre National de la Recherche Scientifique, Université Paul Sabatier, and by a fellowship 'Fin de thèse' from the Fondation pour la Recherche Médicale (FDT202204014744 to D.J.).

## Author contributions

N.C. and R.C.-L. conceived the project. D.J., A.L., I.M-B. and N.C. performed experiments. D.J generated fluorescent strains, performed fluorescence microscopy experiments (TIRFM and SIM$^2$) and Y2H experiments. A.L. assisted in SIM$^2$ experiments. I.M-B. constructed the plasmids for Y2H experiments, generated strains and performed split luciferase complementation assays and western blot analyses. N.C. generated overexpression and fluorescent pneumococcal strains, and performed protein complex purifications and western blot analyses. D.J. C.B., A.B. and A.L. processed, analyzed and interpreted microscopy data. C.B. established the TIRFM workflow, circular kymographs and assisted in general microscopy data collection and analysis. A.B. wrote the python scripts that measured doubling time and cell morphology in microscopy time-lapses. P.P/N.C. and R.C.-L. provided infrastructure and scientific advice. R.C.-L. provided the TIRFM and SIM microscopes. D.J., N.C. and R.C.-L. wrote the manuscript.

## Competing interests

The authors declare no competing interests.
