## [Transparent Peer Review file · Nature Communications]

Transient inhibition of cell division in competent pneumococcal cells results from deceleration of the septal peptidoglycan complex

Corresponding Author: Dr Nathalie Campo

Version 0:

Reviewer comments:

Reviewer #1

(Remarks to the Author)

The paper “Transient inhibition of cell division in competent pneumococcal cells results from deceleration of the septal peptidoglycan complex” is thorough and well-written. This paper gives a important and novel insight into how division is slowed during *Streptococcus pneumoniae* competence, showing that the association of ComM with the divisome somehow reduces the divisome speed. I believe this paper should be published in Nature communications after a few points are addressed.

1. One main point I hope to see addressed is the movies supplied with the manuscript, as this is where I and other readers go to see the motion to evaluate the input data that went into the kymographs, and without being able to visually resolve the directional movements of particles the kymographs are in doubt. (stated as kymographs – which can get the velocity of a particle - should always be drawn along the line of a particle showing visible motion).

I raise this point as it is incredibly hard to see any “directional motions” of the spots in most (but not all) movies, making me wonder what moving particles are they seeing that they are basing their claim of directional motion. This should be remedied to back up their claims of directional motion.

1a. First, the authors should put markers into each movie pointing out particles that show directional motion, helping the reader see and evaluate the dynamics. This could be done in 2 ways, either 1) adding arrows into the movie noting where the particle showing directional motion is, or better yet, 2) tracking the particles with trackmate, and showing the tracked particle, and overall trace that emerges. (trackmate has an option to export movies).

1b. Finally, these movies are incredibly small, making it hard to evaluate the motion of particles. Moreover, if one zooms into the movies the images sufferer from anti-aliasing effects (a problem of all operating systems) as the pizel edges become blurs Thus, would also greatly help the reader if these movies were made larger in size - “scaled up”. A simple way to do this in Fiji – and avoid anti-aliasing artifacts - is to use the “scale” function with “no interpolation”.

1c. Finally, as the entire paper hinges on the directional movement of ComM, this should be visible in the attached movies and demonstrated multiple times from a few different timelapses (currently I can only see directionally moving particle in Movie 1, and cannot resolve any in movie 2). Thus, the authors should add (or append to movies 1 and 2) 1-2 more examples (timelapses) of different cells showing this directional motion.

2. A few notes regarding the statement “The same speed was measured independently in vertically oriented cells from kymographs generated around the cell circumference” and Extended data 4B.

2a. Extended Data Fig. 4’s title is “ComM nodes move in both directions around the cell circumference”, a point not mentioned in the text, and one I find highly questionable given the kymograph shown in 4B (see below).

2b Extended Data 4c - The number of traces used to get this measurement for is 16, far too small of a number to get statistics from. This number should be increased if they want to make this claim.

2c. The kymograph data (Extended data 4b) used to make this claim is incredibly noisy, and their drawn lines are extremely questionable given the supplied data in this figure. Velocities taken from kymographs (measuring the slopes) are well known to be notoriously error-prone, even more so when extrapolated from very noisy data. This is definitely the case for this data.

In contrast to all the other kymographs in this paper, this kymograph is extremely noisy, with extremely undefined edges to draw angles along, leading highly questionable lines drawn along these potential diagonals. Many assigned lines (4B right) are not visible at all in the kymograph (for example, the upper right line), and many of the other lines appear to be arbitrary, trying to fit extremely noisy slopes (like the bottom right line, and the middle-left line). Overall, the kymograph / velocity data in this figure appears highly questionable and, in my opinion, this data hurts the paper more than it helps, as it might cause readers to question the other kymograph data, if it is equally as noisy and unclear.

Overall, regarding Extended figure 4 – It would take a great deal more data collection any kymographs to make the claim of the “same velocity” when measured around the circumference in a convincing manner. However, in this reviewer’s opinion – the authors do not need this figure, or need to make the claim of similar rates of motion. Overall it the paper 1) They have enough compelling data gaining the velocities from horizontal cells, roughly making their point about motion, 2) none of the rest of the velocity measurements of other proteins. are done in this orientation, and 3) a circumferential velocity measurement does not add much to the paper at all, nor change any of the conclusions.

As increasing the number of trajectories and showing clearer kymographs for the circumferential motion could be a lot of work, this reviewer suggests it might be far easier to remove Extended data 4B and the statement that the circumferential motion occurs at the same speed in the text.

3. The use of “Nodes” in colocalization study in Figure 4 is confusing. The paper appears to use “nodes” as a roughly defined descriptor of large bits of signal that they can see move in Figure 1. In Figure 4, they work to determine the colocalization of ComM and PBP2x, at first quantating the overlap of “nodes”. The node analysis is incredibly unclear, as what comprises a “node” of signal is entirely undefined. Looking at the data non-overlapped data in Extended Data 10 shows not discrete clusters of signal, but rather long arcs and smears with some increases in intensity (With the exception panel 2 E, those images show more discrete data).

3a. Out of these arcs and smears, how do the authors define and segment out “nodes”? If they keep this analysis, the processing or thresholding they use to define nodes should be clearly defined. What classifies as a “node” as it appears arbitrary.

3b. Moreover, why is defining nodes necessary here? Their quantitation of ComM/PBP2x overlap in Figure 4F already makes this point clear, much more than the undefined node analysis. I suggest removing the “node” analysis (unless way better defined), as the colocalization data already makes the point (and the fact the see the spots move in 2D).

3c. If this data remains. One more thing that should be reported the overall time it took to take these 2 color 3D-SIM images. This overall time for each image should be reported as taking SIM images at multiple Z-planes takes on more than 10 seconds in this reviewer’s experience (on the Elyria 7). I note this as, during the acquisition, a great some of the signal of moving nodes or diffusive protein may have moved, and thus appear more smeared arcs with their increased resolution.

Minor points.

4. Suggestion - In the discussion, they note this fits with a model that ComM might slow the divisome complex by interacting with DivIB to prevent activation of FtsW:PBP2x. This is very confusing, if ComM prevents activation of FtsW:PBP2x, then the divisome should not move, but here the complex is slowed - a reduction in activity) – and is still “activated” as it moves. Thus, it is far more likely that ComM is just reducing the activity of the Divisome, not turning it on or off through activation.

5. Suggestion - Regarding ComM's putative protease activity, they note "this raises the intriguing possibility that ComM might specifically degrade DivIB as substrate". If the complex is degraded, the divisome will stop; they give no explanation why DivIB degradation could slow the complex and why it would not just stop or fall apart.

6. The halo tag labeling in this paper is unclear and not defined in the text or methods. Did they label PBP2x fully or sparsely? What dye was used, what concentration was used, and for how long were the cells treated, and were the cells washed afterwards.

7. N- values are missing from figure 3.

8. The statement on lines 74-76 is not fully accurate, as it is different in Bacillus. "The model that has emerged is that FtsZ treadmilling is not rate-limiting for septum constriction and does not control the processive movement of active septal PG synthases as originally suggested"

While it is true the treadmilling rate of staph and E. coli is not limiting for septum constriction, it should be noted that all studies so far have found treadmilling limits the rate of septum constriction in B. subtilis. Granted, the motion of the enzymes and treadmilling in B. subtilis is uncoupled (they move at different rates) as shown by Whitley et al correcting the mistake in Bisson et al. However, every paper examining the relationship of treadmilling and septum constriction in bacillus still shows that the rate of treadmilling does indeed control the rate of septum construction.

For example, the figure titled "Fig. 4: Constriction rate is accelerated by FtsZ treadmilling and fast cell growth rate" in the Whitley 2021" demonstrates that after condensation, ring constriction is slowed by PC19. They show in "Supplementary Figure 16: Effect of FtsZ(D213A) expression on treadmilling speed" that expression of D213A also greatly slows constriction. This data confirms the findings in Bisson et al paper - while they were incorrect about the enzyme motion being coupled ("or driven") to treadmilling - they also observed the constriction rate to decrease (and even increase) with they adjusting the rate of treadmilling.

Ethan Garner

(Remarks on code availability)

Reviewer #2

(Remarks to the Author)

Cell division is of fundamental interest and involves a set of conserved proteins across the bacteria. Despite this importance we still do not understand how the protein network functions to allow morphogenesis to occur resulting in two daughter cells. Cell division must be highly regulated and during competence in S. pneumoniae division is temporarily inhibited to allow efficient DNA acquisition. ComM is the division inhibitor and the manuscript by Juillot et al. has begun to determine the mechanism by which it exerts its effect. Several different, complementary approaches have been undertaken, which have demonstrated that ComM slows the speed of the septal peptidoglycan biosynthetic machinery and seems to do this by interaction with the regulatory component, DivIB. Overall, a lot of nice work is presented, and the manuscript makes a very useful contribution.

We have several points to be addressed:

1. Extended Data Fig. 1. The loading controls for the Western blots are missing.
 2. Is the ComM fusion fully functional as stated (l. 123-125). Compare yellow violins in ED Fig. 2, CSP+ WT and CSP+ mNG-comM. Not the same negative controls but if they were measured as relative change this would allow their comparison.
 3. Fig. 1 legend says, 'just before imaging'. From the M&Ms, the doubling time is measured from 5 min after CSP addition and then only for cells that have obviously finished one round and initiated another (therefore only new rounds are timed?). This leads to a question for all the protein dynamics studies. Does ComM slow the rate of all dividing cells or only those that have initiated a new round after its induction? This maybe important as ComM maybe be able to insert and inhibit an existing process or has to be there from the beginning.
 4. As ComM has a role in competence, at what timepoint and in what cells (all or just new slow dividing) do they actually take up exogenous DNA? This could be tested by using labelled DNA for uptake, perhaps.
 5. Why does ComM slow division? Presumably the organism coordinates chromosome replication and separation with septation and so does the decrease in septum synthesis rate permit resolution of chromosome separation? What happens to chromosome separation in the presence of ComM (DAPI staining?)? I presume this is already published.
6. l.242-244. A GFP-PBP2b control with methicillin +/- CSP would strengthen the assertion that the methicillin effect is

mostly specific to PBP2x.

7. Interestingly ComM and PBP2x only partially colocalise ComM (Fig. 4). Why would this be so? Given the size of the nodes (arcs) for each protein around the circumference in Fig 4D, is the level of overlap more than one would expect from a random node localisation?

8. Fig. 4: Perhaps the RodA colocalization is not the best control as this is not involved in septum formation. Does ComM colocalise with DivIB as would be predicted by the Y2H etc?

9. What are the proposed topologies of DivIB and ComM? A diagram would be really useful for the reader. ComM and FtsW interact by split luciferase, which is very interesting. In Fig. 6C, do FtsW and ComM interact as would be predicted from Fig. 6B? Does induction of competence reduce the level of PBP2x - FtsW interaction in strain R5303 (using the approach in Fig. 6C).

10. Does overexpression of FtsW suppress ComM effects on division rate?

11. Line 199 and ED Fig. 8. When the authors refer to "higher background relative to FtsZ", it is not immediately obvious when looking at the figure (as there is no FtsZ data there). Could this be clarified?

12. Extended Data Fig. 10A. First panel should be named "Merged".

13. Line 347, please correct the typo from DivIVB to DivIB.

14. The Discussion is a bit long in places and summarises the results (e.g., l. 385-394).

15. The model in Fig. 7 needs to consider the apparent ComM - FtsW interaction.

16. l. 413-415. Contrary to the suggestion, there is some characterisation of the role of Abi proteins in bacteria which may suggest their function in cell division and peptidoglycan metabolism. For example: Schaefer et al. (2021) Nature Microbiology. 6. 1-10. 10.1038/s41564-020-00808-5; Willing et al. (2021) J Bacteriol 203:10.1128/jb.00014-21.

17. l. 437-442 The above findings for an Abi protein controlling peptidoglycan hydrolase activity might be worth mentioning here?

18. Line 474, and Fig 5 etc. For those readers not familiar with the system, it would be useful to know what BIP is.

19. The reference list does not always use italics for the scientific names of bacteria.

(Remarks on code availability)

Reviewer #3

(Remarks to the Author)

(Remarks on code availability)

Reviewer #4

(Remarks to the Author)

Review for Juillot et al. 'Transient inhibition of cell division in competent pneumococcal cells results from deceleration of the septal peptidoglycan complex'

This submitted manuscript provides significant insights into the mechanisms of how a competence-induced protein ComM interferes with cell division. This manuscript uses state-of-the-art microscopy techniques to study the movements of PG synthesis machinery in relationship to competence development. In combination with genetic screens and biochemical characterization, the authors present a model for the regulation of septal PG synthesis by ComM via DivIB.

A previous report showed that ComM, a membrane protein induced during pneumococcal competence, inhibits cell division. This current paper demonstrates that the induced ComM moves together with the FtsW:PBP2x septal synthesis complex, and reduces the speed of this complex. The interference with PG synthesis is specific to the septal complex, while the movement of the peripheral complex is not affected. A genome-wide Y2H screen identified DivIB as a potential interactor with ComM, which was subsequently confirmed with split-luciferase complementation assays and co-immunoprecipitation experiments. The findings are interesting in terms of general mechanisms of PG machinery movements in response to

internal or external stress.

The manuscript is carefully and clearly written. The rationales are clearly stated and the quality of the data is high. I appreciate the detailed description of strain constructions and the different assays. The references are complete and appropriate. Their lines of work will further the use of examination of molecule movements as an assay for physiological phenomenon.

Major points

1. Include DivIB-mNG in the analysis performed in extended data fig. 1A. This will also answer the question whether the addition of CSP changes DivIB amounts. Line 418-421 'FtsW and PBP2x protein levels were comparable in competent and non-competent cells (Extended Data Fig. 1A), indicating that ComM does not affect the amount of active septal PG synthase complexes. However, this raises the intriguing possibility that ComM might specifically degrade DivIB as substrate.' The authors therefore must have questioned if DivIB amounts change in response to CSP addition. DivIB-mNG was detected in the membrane lysate fraction in Fig.6C. If DivIB is below the detection level using the whole cell extract, membrane lysate fractions may be necessary.
2. Addition of a summary table or graph to show the velocities of different molecules under different conditions. Include the full name of the fusion protein (e.g. mNG-comM, FtsZ-mNG etc) and conditions (competent cells, noncompetent cell; methicillin addition; inducer addition etc). This will be very useful for cross-referencing.
3. Co-immunoprecipitation experiment in Fig. 6C: This figure should include negative control strains that do not have mNG tag (for sample, untagged PBP2x FtsW-ALFA strain and untagged DivIB ALFA-ComM strain). This is to control for nonspecific binding of proteins to the mNeogreen trap. mNG-DivIB label should be DivIB-mNG.
4. Include MW bands in the blot. The section of the blot shown should include the area with one MW size above and one MW size below. It will be useful to add a full-size blot in the supplemental figure. Put the labels of the antibody on the side of the panels so that it is clear that left and middle panels are labeled with anti-mNG and anti-ALFA. Label the mNG proteins as 'bait' and the ALFA proteins as 'prey'. Mention the numbers of biological replicates performed.

Minor points

Figures:

5. Extended Data Fig. 1: Mention in the figure legend the expected size of mNG, untagged protein and the fusion proteins. To demonstrate that the lower band obtained with anti-HT is a nonspecific band, include a lane obtained with a WT sample. Alternatively, use an anti-HaloTag monoclonal antibody (Promega, G921A, 1:1000), which do not give non-specific bands.
6. Some of the labeling and lines on the graphs (Fig. 1 H, 1I, 2B, 2C, 2F, 3, 5, 6D and 6G). are very light. Change the light orange, light blue or yellow to a deeper shade.
7. Mention the numbers of biological replicates performed for experiments shown in the figures. For example, mention the number of biological replicates in figure 1H. For figure 1I, are the values for each time point obtained from one experiment or from multiple experiments? The same clarifications apply to Fig. 2C, 2D and others.
8. For other experiments that show representative results (fig. 1B, 1C, 1E, 1F, 2A etc), provide the number of biological replicates performed for each kind of experiment.
9. Legend to Figure 2B and 2F, mention that the color scheme in 2F corresponds to the cells with particular Z-ring profile.
10. Figure 5, need to clarify BIP (see comments on Materials and Methods). Label mNG-pbp2x movement on the graph.
11. Legend to figure 6 'Whole cell extracts prior to immunoprecipitation (L, load)', is L the whole cell extracts, or is it the membrane extract? Add volume loaded for each lane. Please also add volume used for the buffers used for the membrane preps and resuspension in the materials and methods section.

Text:

12. Line 74-76: 'The model that has emerged is that FtsZ treadmilling is not rate-limiting for septum constriction and does not control the processive movement of active septal PG synthases as originally suggested.' Put in references after 'septal PG synthases' and put in reference after 'originally suggested'
13. Line 97-98: 'ComM localizes at midcell and does not affect assembly and localization of divisome components, suggesting that it transiently inhibits the active process of constriction.' Provide a reference for this sentence or mention that this is shown in this study.
14. Line 130: mention that microscopy was performed at 37°C because other groups perform microscopy at different temperatures. Add a reference for HILO microscopy.
15. Line 177: perhaps add 'but not the components of the peripheral PG synthase complex' to emphasize the effect is specific to the septal PG synthase complex.
16. Line 292: PBP2x displayed the same average speed than in wild-type cells in the absence...Change 'than' to 'as'.
17. Line 323, also change 'than' to 'as'
18. Lines 331 to 332: MurA is mentioned in this manuscript only once here. Give a one-sentence background on MurA and MurZ. 'MurA and MurZ are the two homologues of the MurA family in pneumococcus that catalyze the first committed step of PG synthesis.' Give a spr # after MurA because the identities of the two homologues can be confusing.
19. Different sections of Discussion: put in figure # so readers can refer to the figures. For example, line 371 'Our findings indicate that the transient inhibition of cell division that occurs during competence in *S. pneumoniae* results from ComM-dependent deceleration of FtsW:PBP2x (Fig. 3A to 3C), etc.
20. Line 407, *S. pneumonia* should be *S. pneumoniae*.
21. Line 413-414, reference also 43, Straume et al, 2017.
22. Line 457, no supplementary for Table 1 and 2.
23. Line 474, the references 86 and 87 did not use the term BIP-1. Please specify.
24. Line 527, construction of R4846 (Δ comC, mNeonGreen-pbp2x, CEPr-comM (kan)) is missing.
25. Line 678 to 680, include the volume of lysate added to each lane in the figure legend or in the Materials and Methods. 'Gels were activated by ultraviolet (UV) exposure for 45 s using a Bio-Rad ChemiDoc MP imager to visualize and estimate

total protein per lane.' What did you do with the estimation? Did you try to do quantitation and normalize the relative amounts of Pbp2x etc with or without CSP addition?

26. Line 710, spell out TIRF and HILO

27. Lines 852 to 872, include volume for each step.

28. Table 1: what is CEPr in R4846 Δ comC, mNeonGreen-pbp2x, CEPr-comM (kan)?

29. R5323 ((mNeonGreen-ComM, MurA-ALFA) is not listed. Strain construction of this strain should be included in Materials and Methods.

(Remarks on code availability)

I am not familiar with codes.

Version 1:

Reviewer comments:

Reviewer #1

(Remarks to the Author)

The authors have addressed all my concerns, and I strongly believe this manuscript should be published. All authors on this work should be congratulated for this great paper.

(Remarks on code availability)

Reviewer #2

(Remarks to the Author)

The authors have done a thorough job answering and rebutting my comments. The revised manuscript is an important contribution.

(Remarks on code availability)

Reviewer #3

(Remarks to the Author)

(Remarks on code availability)

Reviewer #4

(Remarks to the Author)

I appreciate very much the detailed revision the authors assembled for the manuscript.

I agree with all the authors' changes, and really appreciate the explanations. I look forward to seeing the final version. It is an important contribution to the area.

For the section on immunoprecipitation, I appreciate that the equivalence of the volume of the original culture is specified. I always wondered about the relative amounts of pull down in other publications.

I just have a few minor comments and suggestions

Figure 6: add 'bait' under mNG-DivIB, and 'Prey' under ALFA-ComM. It also will be useful if R5300 (mNG-DivIB ALFA-Com) and R5298(ALFA-ComM) are added to the figure. It will make it easier to follow the figure.

Supplementary Figure 3: Please add whether the cultures were treated with csp for a certain duration.

Supplementary Figure 11: please also add the labels 'bait' and 'prey' to this figure.

Main text, under materials and methods

Whole-cell extracts preparation and immunoblot analysis. Line 707: 'Samples (3 ml) were collected by' ...is different from line 713 to 714 'For immunoblot analysis, we loaded 10 μ l of whole cell extracts resulting from cultures treated with CSP (corresponding to ~600 ml of cells). The cell volumes are different in the two sentences.

Line 919: mixed with mNeonGreen-Trap Magnetic Agarose beads (Chromotek) and rotated for 6 h at 4 °C. Can the incubation be longer than 6 h? It will take all day to get the preparation to this step. I assume it will be easier if the experimenter can go home and come back the next day to finish the preparation. Please give a range of time (6 to 12 or 6 to 18 , for example) that would work for this method.

A minor comment: supplementary figure 1a, anti-HT: As suggested to the authors, the expected sizes of the fusion proteins were added in the figure legend. The authors also switched to a commercial antibody that does not give nonspecific band. However, the figure still has one issue. RodA-HT MW is expected to be 81 kD. but the band shown is about 60 kD. A

negative control (WT) with a strain with no HT tag will clarify that the band is not non-specific. However, this point is shown by the absence of extra bands in the ht-pbp2x samples. Since anti-rodA is not available, it is not possible to determine whether rodA-HT is degraded. It is not necessary to add a non-tag strain in this case because of the pbp2x samples. However, I want to re-emphasize the need to validate all antibodies (commercial or home-made) for their specificity.
Ho-Ching Tiffany Tsui

(Remarks on code availability)

RESPONSES TO REVIEWERS COMMENTS

[manuscript # NCOMMS-24-65120-T, by Juillot *et al.*]

We thank the three reviewers for their insightful comments. Our revision addresses all their points, with some analyses and additional data extending beyond their requests. We have generated new data through additional experiments and reanalyzed nearly all our imaging data with greater accuracy. We believe our manuscript has been significantly improved and data become even more robust - we thank the reviewers again for their valuable input throughout this process.

In summary, our main changes and additions are as follows:

- Regarding our measurements of speeds (Fig. 1,2,3,5,6 and related Supplementary Figures):
 - **The data used to generate all histograms of speeds distributions over time have been revised and homogenized.**
 - We realized that some speed distributions included data from cells imaged > 30 minutes after CSP induction, which does not accurately reflect the behavior of competent cells. These have been removed.
 - Most of the graphs reporting speeds in the original manuscript displayed results from a single experiment representative of two or three independent replicates. In the revised manuscript, all figures include the data from all replicates, reducing variability and ensuring greater consistency.
 - **We added a new supplementary Table** that compiles the speeds of all molecules across all conditions analyzed in our study
 - **New acquisitions were performed on cells in vertical chambers to increase the number of trajectories to measure ComM velocity** around the cell circumference, and we also treated the movies to reduce background noise and generate better kymographs
 - **All accompanying videos have been re-edited to help the reader** to observe and assess the dynamics of particles within division rings. Each video now includes an average projection and the kymograph associated to a division ring.
- Regarding our colocalization analyses from SIM images (Fig. 4):
 - **We confirmed the alignment of the two channels using bi-color beads.**
 - **We performed the Pearson correlation coefficient (PCC) analysis in the original SIM images, instead of on SIM² reconstructed images**, in which the signal intensity might be non-linear relative to the raw data. SIM-based PCC analysis is more rigorous.
 - **We added additional controls** including strains expressing single fluorescent fusions
 - **We added colocalization analysis of ComM:DivIB**, absent from the original manuscript.
 - **The node analysis has been removed** following the suggestion of reviewer #1
- Regarding the co-immunoprecipitation experiments (Fig. 6 and S11):
 - **We added negative control strains** to test the interaction between DivIB and ComM.
 - **We tested ComM:FtsW interaction** as suggested by reviewer #2, and also ComM:PBP2x
 - During the course of the revision of the manuscript, we found that the FtsW-ALFA fusion that we used as a positive control in combination with the mNeonGreen-PBP2x fusion, was detected in the untagged control immunoprecipitates. We therefore did new immunoprecipitation experiments with strains containing instead the FtsW-HaloTag fusion (and mNeonGreen-PBP2x) and the FtsW-mNeonGreen fusion (and HaloTag-PBP2x)
- **We produced new immunoblot analyses** of the strain containing the mNeonGreen-DivIB fusion, as recommended by reviewers #2 and 4, but also of the strain containing the RodA-HaloTag and mNeonGreen-ComM fusions
- **We analysed the stability of DivIB during the course of competence** in wild-type and $\Delta comM$ mutant cells by immunoblot, to assess a possible protease activity of ComM.

Please find below our detailed point-by-point response (in blue) to the reviewers' comments.

Reviewer #1 (Remarks to the Author):

The paper “Transient inhibition of cell division in competent pneumococcal cells results from deceleration of the septal peptidoglycan complex” is thorough and well-written. This paper gives a important and novel insight into how division is slowed during *Streptococcus pneumoniae* competence, showing that the association of ComM with the divisome somehow reduces the divisome speed. I believe this paper should be published in Nature communications after a few points are addressed.

1. One main point I hope to see addressed is the movies supplied with the manuscript, as this is where I and other readers go to see the motion to evaluate the input data that went into the kymographs, and without being able to visually resolve the directional movements of particles the kymographs are in doubt. (stated as kymographs – which can get the velocity of a particle - should always be drawn along the line of a particle showing visible motion).

I raise this point as it is incredibly hard to see any “directional motions” of the spots in most (but not all) movies, making me wonder what moving particles are they seeing that they are basing their claim of directional motion. This should be remedied to back up their claims of directional motion.

We thank the reviewer for highlighting this area for enhancement. We have considered this feedback and provide updated versions of the films that are more comprehensible for the reader and rectifies the issues that emerged during the data export process. The technical specifics can be found in the reviewer's recommendations provided below.

1a. First, the authors should put markers into each movie pointing out particles that show directional motion, helping the reader see and evaluate the dynamics. This could be done in 2 ways, either 1) adding arrows into the movie noting where the particle showing directional motion is, or better yet, 2) tracking the particles with trackmate, and showing the tracked particle, and overall trace that emerges. (trackmate has an option to export movies).

We are grateful for this suggestion. All movies have been edited to facilitate the reader's analysis of the dynamics of the moving particles. The final video now comprises three panels containing, arranged from left to right: the image time sequence - an average projection - the kymograph generated on the particle pointed by an arrow. The time-lapse sequence has also been pre-processed (subtraction by the minimum projection) to emphasize the signals of interest. A mobile cursor traverses the kymograph, thereby establishing a correlation between the movement of a protein on the film and its representation in the kymograph. We hope that this revised presentation will facilitate a more comprehensive evaluation by the reader.

1b. Finally, these movies are incredibly small, making it hard to evaluate the motion of particles. Moreover, if one zooms into the movies the images sufferer from anti-aliasing effects (a problem of all operating systems) as the pizel edges become blurs Thus, would also greatly help the reader if these movies were made larger in size - “scaled up”. A simple way to do this in Fiji – and avoid anti-aliasing artifacts - is to use the “scale” function with “no interpolation”.

We apologize for the poor quality of our original films, which was degraded when the files were converted to AVI format. As suggested, the original data has been scaled up by a factor of 4 without interpolation, resulting in good quality AVI files.

1c. Finally, as the entire paper hinges on the directional movement of ComM, this should be visible in the attached movies and demonstrated multiple times from a few different timelapses (currently I can only see directionally moving particle in Movie 1, and cannot resolve any in movie 2). Thus, the

authors should add (or append to movies 1 and 2) 1-2 more examples (timelapses) of different cells showing this directional motion.

As mentioned in section 1a, for each of the fluorescent fusions studied, the new edited movie displays both the initial temporal sequence and a kymograph. This new representation makes it possible to establish a link between the directional movements of the film and their representation on the kymograph.

2. A few notes regarding the statement “The same speed was measured independently in vertically oriented cells from kymographs generated around the cell circumference” and Extended data 4B.

2a. Extended Data Fig. 4’s title is “ComM nodes move in both directions around the cell circumference”, a point not mentioned in the text, and one I find highly questionable given the kymograph shown in 4B (see below).

2b. Extended Data 4c - The number of traces used to get this measurement for is 16, far too small of a number to get statistics from. This number should be increased if they want to make this claim.

2c. The kymograph data (Extended data 4b) used to make this claim is incredibly noisy, and their drawn lines are extremely questionable given the supplied data in this figure. Velocities taken from kymographs (measuring the slopes) are well known to be notoriously error-prone, even more so when extrapolated from very noisy data. This is definitely the case for this data.

In contrast to all the other kymographs in this paper, this kymograph is extremely noisy, with extremely undefined edges to draw angles along, leading highly questionable lines drawn along these potential diagonals. Many assigned lines (4B right) are not visible at all in the kymograph (for example, the upper right line), and many of the other lines appear to be arbitrary, trying to fit extremely noisy slopes (like the bottom right line, and the middle-left line). Overall, the kymograph / velocity data in this figure appears highly questionable and, in my opinion, this data hurts the paper more than it helps, as it might cause readers to question the other kymograph data, if it is equally as noisy and unclear.

Overall, regarding Extended figure 4 – It would take a great deal more data collection any kymographs to make the claim of the “same velocity” when measured around the circumference in a convincing manner. However, in this reviewer’s opinion – the authors do not need this figure, or need to make the claim of similar rates of motion. Overall it the paper 1) They have enough compelling data gaining the velocities from horizontal cells, roughly making their point about motion, 2) none of the rest of the velocity measurements of other proteins. are done in this orientation, and 3) a circumferential velocity measurement does not add much to the paper at all, nor change any of the conclusions.

As increasing the number of trajectories and showing clearer kymographs for the circumferential motion could be a lot of work, this reviewer suggests it might be far easier to remove Extended data 4B and the statement that the circumferential motion occurs at the same speed in the text.

Thank you for these comments. We have performed an additional experiment and reached a total of 51 trajectories (new Supplementary Fig. 4c). This number may still be weak for solid statistical analyses, but the resulting average speed is consistent with the value obtained from trajectories analyzed in cells lying horizontally (12.2 ± 5.3 nm/s in Fig. 1h and 9.9 ± 2.8 nm/s in Supplementary Fig. 4c). As mentioned above, all movies, including movies obtained from cells in vertical chambers were treated to reduce the background noise. We have replaced the original kymograph shown in Supplementary Fig. 4 with a new kymograph that shows trajectories with different directions, suggestive of bi-directional motion. While we agree with the reviewer that Supplementary Fig. 4 is dispensable, we prefer to keep this complementary analysis of ComM motion.

3. The use of “Nodes” in colocalization study in Figure 4 is confusing. The paper appears to use “nodes” as a roughly defined descriptor of large bits of signal that they can see move in Figure 1. In Figure 4, they work to determine the colocalization of ComM and PBP2x, at first quantating the overlap of “nodes”. The node analysis is incredibly unclear, as what comprises a “node” of signal is entirely undefined. Looking at the data non-overlapped data in Extended Data 10 shows not discrete clusters of signal, but rather long arcs and smears with some increases in intensity (With the exception panel 2 E, those images show more discrete data).

3a. Out of these arcs and smears, how do the authors define and segment out “nodes”? If they keep this analysis, the processing or thresholding they use to define nodes should be clearly defined. What classifies as a “node” as it appears arbitrary.

3b. Moreover, why is defining nodes necessary here? Their quantitation of ComM/PBP2x overlap in Figure 4F already makes this point clear, much more than the undefined node analysis. I suggest removing the “node” analysis (unless way better defined), as the colocalization data already makes the point (and the fact the see the spots move in 2D).

Following the suggestion of reviewer #1, the node analysis we initially proposed in Fig. 4 has been removed. We agree that this complementary analysis was unnecessary -the PCC analysis already makes the point- and the use of ‘nodes’ confusing since the structures formed by ComM signal are difficult to define (indeed forming arcs and smear), and thus they were segmented manually, making their classification a bit arbitrary.

Furthermore, the analysis by colocalization using Pearson correlation coefficients (PCC) has been entirely redone, reaching the same conclusions. In the original version of our manuscript, the analysis was carried out directly on reconstructed SIM² data, which generates reconstructed images with truncated intensities for the lowest values, and sometimes without preserving the linearity of the signal according to the raw data. The images reconstructed using the SIM² approach have been retained solely for the purposes of visualization (Figures 4 d,e,f) and for measuring the diameter of the rings (Figures 4 h,i,j,k). For the PCC colocalization analysis, the images were reconstructed using the classic SIM approach (without intensity clipping and by keeping the linearity of the signal to the raw data). The values obtained are very close to those obtained previously, as can be seen from the graph below.

We have updated Fig. 4g with the PCC of SIM images, which is more rigorous, and we have also added in this figure (a) the controls of single-labelled ComM and PBP2x strains and (b) the PCC analysis of the ComM:DivIB pair

3c. If this data remains. One more thing that should be reported the overall time it took to take these 2 color 3D-SIM images. This overall time for each image should be reported as taking SIM images at multiple Z-planes takes on more than 10 seconds in this reviewer's experience (on the Elyria 7). I note this as, during the acquisition, a great some of the signal of moving nodes or diffusive protein may have moved, and thus appear more smeared arcs with their increased resolution.

Please note that these were 2 color 2D-SIM images (not 3D-SIM) of the division plane in cells immobilized vertically in the chambers. Our acquisitions used 15 phases at 20 ms each (i.e. 3 seconds per channel), which remained compatible with the dynamics of the proteins reported here (speed ~ 10-40 nm/s).

Minor points.

4. Suggestion - In the discussion, they note this fits with a model that ComM might slow the divisome complex by interacting with DivIB to prevent activation of FtsW:PBP2x. This is very confusing, if ComM prevents activation of FtsW:PBP2x, then the divisome should not move, but here the complex is slowed - a reduction in activity) – and is still “activated” as it moves. Thus, it is far more likely that ComM is just reducing the activity of the Divisome, not turning it on or off through activation.

We apologize if our phrasing was confusing. We did not mean that ComM inactivates the septal PG complexes in an ‘on-off’ manner but possibly that it reduces their activity. This has been corrected in the text (lines 418-421).

5. Suggestion - Regarding ComM's putative protease activity, they note “this raises the intriguing possibility that ComM might specifically degrade DivIB as substrate”. If the complex is degraded, the divisome will stop; they give no explanation why DivIB degradation could slow the complex and why it would not just stop or fall apart.

This is a valid point. However, we have now analyzed the level of the mNeonGreen-DivIB fusion at different time after competence induction (new Supplementray Fig. 13) and found that the protein remains stable and at similar levels in the presence and in the absence of ComM. Thus, it is unlikely that ComM degrades DivIB.

6. The halo tag labeling in this paper is unclear and not defined in the text or methods. Did they label PBP2x fully or sparsely? What dye was used, what concentration was used, and for how long were the cells treated, and were the cells washed afterwards.

We apologize for this omission. We have added this information in the Material and Methods section of the revised manuscript (lines 434 to 737) as follows:

“To visualize Halo-Tag fusions, Janelia Fluor® 549 HaloTag® ligand (GA1110, Promega) was added to cells at a final concentration of 20 nM, and incubated for 10 minutes with no washing step before imaging.”

7. N- values are missing from figure 3.

We apologize for this omission. N-values are now indicated in the legend. Note that in the original manuscript, most of the graph were generated with data from only one of two independent replicates. All figures now include the data from all the replicates, thereby attenuating experimental variations.

8. The statement on lines 74-76 is not fully accurate, as it is different in Bacillus.

“The model that has emerged is that FtsZ treadmilling is not rate-limiting for septum constriction and does not control the processive movement of active septal PG synthases as originally suggested”

While it is true the treadmilling rate of staph and E. coli is not limiting for septum constriction, it should be noted that all studies so far have found treadmilling limits the rate of septum constriction in B. subtilis. Granted, the motion of the enzymes and treadmilling in B. subtilis is uncoupled (they move at different rates) as shown by Whitley et al correcting the mistake in Bisson et al. However, every paper examining the relationship of treadmilling and septum constriction in bacillus still shows that the rate of treadmilling does indeed control the rate of septum construction.

For example, the figure titled “Fig. 4: Constriction rate is accelerated by FtsZ treadmilling and fast cell growth rate” in the Whitley 2021” demonstrates that after condensation, ring constriction is slowed by PC19. They show in “Supplementary Figure 16: Effect of FtsZ(D213A) expression on treadmilling speed” that expression of D213A also greatly slows constriction. This data confirms the findings in Bisson et al paper - while they were incorrect about the enzyme motion being coupled (“or driven”) to treadmilling – they also observed the constriction rate to decrease (and even increase) with they adjusting the rate of treadmilling.

Thank you for spotting this and for the explanation – very much appreciated. We have corrected the text accordingly (lines 74 to 76). The sentence reads now:

“The model that has emerged is that FtsZ treadmilling drives Z-ring condensation and affects septum constriction to different extents in different bacteria, but does not control the processive movement of active septal PG synthases as originally suggested²¹.”

Ethan Garner

Reviewer #2 (Remarks to the Author):

Cell division is of fundamental interest and involves a set of conserved proteins across the bacteria. Despite this importance we still do not understand how the protein network functions to allow morphogenesis to occur resulting in two daughter cells. Cell division must be highly regulated and during competence in S. pneumoniae division is temporarily inhibited to allow efficient DNA acquisition. ComM is the division inhibitor and the manuscript by Juillot et al. has begun to determine the mechanism by which it exerts its effect. Several different, complementary approaches have been undertaken, which have demonstrated that ComM slows the speed of the septal peptidoglycan biosynthetic machinery and seems to do this by interaction with the regulatory component, DivIB. Overall, a lot of nice work is presented, and the manuscript makes a very useful contribution.

We thank the reviewers for carefully reviewing our revised manuscript and raising several constructive suggestions.

1. Extended Data Fig. 1. The loading controls for the Western blots are missing.

We used stain free staining as loading control. Stain free gel after activation with UV light are now included in the revised version of Fig. S1. We also added the relative levels of each fluorescent protein fusion for competent versus non-competent cultures in Fig. S1a. Please, see also our response to comment 25 of Reviewer 4.

2. Is the ComM fusion fully functional as stated (l. 123-125). Compare yellow violins in ED Fig. 2, CSP+ WT and CSP+ mNG-comM. Not the same negative controls but if they were measured as relative change this would allow their comparison.

The reviewers make a valid point and we apologize for the confusion. As developed in our response to comment 3 below, doubling times were measured automatically. While this unbiased automatic method allowed the analysis of several hundreds of cells in all conditions and for all strains in our study, it also likely included cells that were already exiting competence, resulting in a slightly faster doubling time for competent cells than we previously reported from manual analysis (see response to comment 3 below, and Bergé *et al.* 2017, doi:10.1038/s41467-017-01716-9; Mortier-Barrière *et al.* 2020, doi:10.3390/genes11060675). We consider, however, that the automated method is a good alternative given that we used it to measure all doubling times, including for control WT cells.

In the original supplementary Fig. 2, data from only one of the two independent replicates were used. The figure now includes the data from all the replicates, thereby attenuating experimental variations. The new supplementary Fig. 2 now clearly shows that the WT strain and strain R4869 containing the mNeonGreen fusion behave similarly in the presence and in the absence of CSP.

3. Fig. 1 legend says, 'just before imaging'. From the M&Ms, the doubling time is measured from 5 min after CSP addition and then only for cells that have obviously finished one round and initiated another (therefore only new rounds are timed?). This leads to a question for all the protein dynamics studies. Does ComM slow the rate of all dividing cells or only those that have initiated a new round after its induction? This may be important as ComM may be able to insert and inhibit an existing process or has to be there from the beginning.

The reviewers raise a valid point here. We chose to automatically measure doubling times of single cells in time-lapse microscopy experiments. This approach allowed to measure the interval between two successive division events initiating within the first 20 minutes following CSP addition - only new division rounds initiated after competence induction were analyzed.

We have previously demonstrated by measuring morphological parameters of single cells (Bergé *et al.* 2017, doi:10.1038/s41467-017-01716-9), and using microfluidics combined with time-lapse fluorescence microscopy (Mortier-Barrière *et al.* 2020, doi:10.3390/genes11060675), that CSP instantly causes a delay of the cell division process in all cells in the population. Production of ComM in the first minutes following competence induction (see Supplementary Fig. 1b), results in a transient inhibition of two key steps in the division process: initiation of constriction in pre-divisional cells and septum closure (Bergé *et al.* 2017). Manually monitoring individual cells lineages further revealed that daughter cells derived from CSP-treated cells recovered doubling times similar to non-competent cells (Mortier-Barrière *et al.* 2020).

4. As ComM has a role in competence, at what timepoint and in what cells (all or just new slow dividing) do they actually take up exogenous DNA? This could be tested by using labelled DNA for uptake, perhaps.

We previously analyzed the binding of labelled DNA fragments to competent cells, as suggested by the reviewers (Bergé *et al.* 2013, doi:10.1371/journal.ppat.1003596, and Kurushima *et al.*, doi:10.7554/eLife.58771). Our findings indicated that transforming DNA can bind nearly every cell in the competent population regardless of their cell cycle state. Moreover, comparison of the transformation frequencies for a PCR DNA fragment carrying a single point mutation conferring streptomycin resistance revealed no significant difference between wildtype and "fast dividing" $\Delta comM$ mutant cells, suggesting that ComM has no impact on DNA uptake (Bergé *et al.* 2017, doi:10.1038/s41467-017-01716-9).

5. Why does ComM slow division? Presumably the organism coordinates chromosome replication and separation with septation and so does the decrease in septum synthesis rate permit resolution

of chromosome separation? What happens to chromosome separation in the presence of ComM (DAPI staining)? I presume this is already published.

Indeed, we previously published that the ComM-dependent cell division delay allows time for completion of transformation, replication, and chromosome segregation. It is particularly important for transformation events that lead to genome rearrangements with the creation of a chromosome dimer as an intermediate (Bergé *et al.* 2017, doi:10.1038/s41467-017-01716-9).

We tried to visualize an increase of guillotined chromosome dimers during transformation using a HlpA(HU)-GFP fusion and DAPI to stain the chromosome, with no success. We estimated that 30% of the cells produce a dimer when transformed with pneumococcal chromosomal DNA (Johnston *et al.* 2015, doi:10.1371/journal.pgen.1004934). Our results suggested that 20% of these cells would be not viable in the absence of ComM (Bergé *et al.* 2017, doi:10.1038/s41467-017-01716-9). However, competent cultures are not synchronized and we don't know how long cells with guillotined chromosome remain intact before lysis. The proportion of guillotined chromosomes could thus be lower than expected in samples harvested at specific time points for microscopy analysis. We were also unable to detect an increase proportion of anucleated cells in these cultures. This could be because daughter chromosomes are mostly segregated in the future daughter cells before the septum is closed. We are currently developing new strategies based on various super-resolution microscopy techniques to analyze chromosome integrity during transformation. These experiments are beyond the scope of this manuscript.

6. I.242-244. A GFP-PBP2b control with methicillin +/- CSP would strengthen the assertion that the methicillin effect is mostly specific to PBP2x.

In *S. pneumoniae*, it is established that methicillin at concentrations ranging from 0.1 to 0.3 µg/ml is primarily selective for PBP2x and to a lesser extent PBP3 (DacA) (Land *et al.* 2013, doi:10.1111/mmi.12408; Kocaoglu *et al.* 2015, doi: 10.1021/acschembio.9b00459). As we noticed that the speed of PBP2x and FtsW slowed down to values similar to the mean speed of ComM in competent cells, our goal was to test whether ComM speed was linked to the speed of the FtsW:PBP2x complex. Thus, we measured the speed of ComM in cells incubated with methicillin, which was previously shown to reduce the movement of PBP2x (Perez *et al.* 2019, doi:10.1073/pnas.1816018116). We chose to use the FtsZ-mNeonGreen strain as a negative control to verify that the methicillin treatment in our conditions had no impact on the physiology of the cell.

7. Interestingly ComM and PBP2x only partially colocalise ComM (Fig. 4). Why would this be so? Given the size of the nodes (arcs) for each protein around the circumference in Fig 4D, is the level of overlap more than one would expect from a random node localisation?

The reviewer makes a fair point about our original 'nodes' colocalization analysis. We agree that this analysis was confusing since the structures formed by ComM signal (patches but also arcs and smears) are difficult to define, and thus the level of colocalization was arbitrarily determined. As per suggestion of reviewer #1, we have removed this analysis since PCC analysis already makes the point. Please see our response to comment 3 of reviewer 1 for more details.

8. Fig. 4: Perhaps the RodA colocalization is not the best control as this is not involved in septum formation.

In Fig. 4, we used strain R5297 containing the mNeongrenn-ComM and RodA-HT fusions as a negative control for the colocalization of mNeonGreen-ComM and HT-PBP2x. In *S. pneumoniae*, both septal (FtsW:PBP2x) and peripheral (RodA:PBP2b) PG synthesis occur at mid-cell, first within the same annular region in pre-divisional cells and later they separate into two concentric regions at

mid-cell (Perez et al. 2021, doi:10.1111/mmi.14659; Trouvé et al. doi:10.1016/j.cub.2021.04.041). Images shown in Fig. 4 suggest that ComM preferentially localizes in the inner septal ring together with the septal PG synthase FtsW:PBP2x, rather than in the outer ring with the peripheral PG synthase PBP2b:RodA. The elongation phenotype of competent cells could result from reduced septation but also from increased elongation vs septation rates and it was important to exclude the latter by determining whether ComM localized in the inner and/or outer rings.

Does ComM colocalise with DivIB as would be predicted by the Y2H etc?

Thanks for this question. We have constructed a new strain (R5389) and shown the colocalization between ComM and DivIB. These new results are shown in Figure 4f,g.

9. What are the proposed topologies of DivIB and ComM? A diagram would be really useful for the reader.

Thanks for this suggestion. A schematic representation of the membrane proteins ComM, DivIB, FtsW and PBP2x has been included in Supplementary Fig. 11a. Models are based on DeepTMHMM prediction.

ComM and FtsW interact by split luciferase, which is very interesting. In Fig. 6C, do FtsW and ComM interact as would be predicted from Fig. 6B? Does induction of competence reduce the level of PBP2x - FtsW interaction in strain R5303 (using the approach in Fig. 6C).

The hypothesis that competence might interfere with the PBP2x:FtsW interaction is attractive. To test this, we did not use strain R5303 producing the mNeonGreen-PBP2x and FtsW-ALFA fusions originally presented in the manuscript, since we found that FtsW-ALFA was detected in the negative control immunoprecipitates (please, see also our response to comment 3 of reviewer 4). We thus performed *in vivo* non-crosslinked co-immunoprecipitation experiments using strain R5386, which contains FtsW-mNeonGreen and HaloTag-PBP2x fusions. However, PBP2x and FtsW failed to co-immunoprecipitate, regardless of whether the cells were induced to develop competence (+CSP), or not (-CSP) (see Figure below, panel A). A possibility to interpret these findings is that the complex FtsW:PBP2x is not sufficiently stable to be detected in the absence of cross-linking.

As suggested by the reviewers, we performed *in vivo* mNeonGreen co-IP experiments to test potential interactions between ComM and FtsW, but also between ComM and PBP2x. For this, we used strains R5385, producing FtsW-mNeonGreen and ALFA-ComM (Figure above, panel B), and strain R4746 producing mNeonGreen-ComM and HaloTag-PBP2x (panel C). Panels B and C above show that we were not able to detect ComM in complex with either FtsW or PBP2x. However, given the absence of co-IP of PBP2x and FtsW in our non-crosslinked conditions, the lack of co-IP of ComM with FtsW or PBP2x remains inconclusive.

These results are now included in the revised manuscript as new Supplementary Fig. 11b and lines 332-337.

10. Does overexpression of FtsW suppress ComM effects on division rate?

Thanks for this suggestion. We have constructed a strain overexpressing FtsW under the control of an IPTG-inducible promoter, but did not observe any difference in growth or cell length compared to the wild-type strain, in competent or non-competent cultures. However, in the absence of pneumococcal anti-FtsW antibodies, it was not possible to verify that FtsW was efficiently overproduced. As we cannot conclude at this stage whether overexpression of FtsW is impacting the effects of ComM on cell division, we decided not to include this data in the revised manuscript.

11. Line 199 and ED Fig. 8. When the authors refer to “higher background relative to FtsZ”, it is not immediately obvious when looking at the figure (as there is no FtsZ data there). Could this be clarified?

We apologize for the confusion; our phrasing was unclear. We have modified the text as follows (lines 198-201):

“...FtsW and PBP2x exhibited diffusive behavior along the membrane (higher background signal in TIRFM summations and kymographs relative to FtsZ) (Compare Supplementary Fig. 7 and 8)...”.

12. Extended Data Fig. 10A. First panel should be named “Merged”.

Corrected - thank you.

13. Line 347, please correct the typo from DivIVB to DivIB.

Corrected - thank you.

14. The Discussion is a bit long in places and summarises the results (e.g., l. 385-394).

We agree that the discussion was a bit too long. We have removed the beginning of the second paragraph that was indeed summarizing the results already presented.

15. The model in Fig. 7 needs to consider the apparent ComM - FtsW interaction.

The model drawn includes a potential interaction between ComM and FtsW. However, we did not emphasize this interaction in the text because we have not been able to validate it by either Y2H or co-IP (please, also see our response to comment 9 above), while all other interactions depicted in the model have been validated.

16. l. 413-415. Contrary to the suggestion, there is some characterisation of the role of Abi proteins in bacteria which may suggest their function in cell division and peptidoglycan metabolism. For example: Schaefer et al. (2021) Nature Microbiology. 6. 1-10. 10.1038/s41564-020-00808-5; Willing et al. (2021) J Bacteriol 203:10.1128/jb.00014-21.

Thank you very much for pointing these articles to our attention. We have modified the paragraph to mention these findings. Please see the revised manuscript, lines 429-337.

17. l. 437-442 The above findings for an Abi protein controlling peptidoglycan hydrolase activity might be worth mentioning here?

In *S. aureus*, the membrane bound domain of the CAAX type II prenyl endopeptidase family protein LyrA is sufficient to support the activity of the SagB PG hydrolase but predicted 'CAAX enzyme' catalytic residues in this domain are dispensable. In contrast, mutations in conserved residues of ComM that could be part of a CAAX-like catalytic site abolished the immunity of competent cells toward the PG hydrolase CbpD, suggestive of a different mechanism and supportive of a protease activity of ComM toward CbpD and/or proteins related to CbpD function that remains to be tested. We have therefore mentioned the above findings for LyrA in support of a regulatory role of DivIB activity through direct protein-protein interaction instead of protease activity (lines 434-437), but not here regarding CbpD (lines 457-463).

18. Line 474, and Fig 5 etc. For those readers not familiar with the system, it would be useful to know what BIP is.

We added the following sentence in the legend of Figure 5:

“BIP-1 (noted BIP on the graph for simplicity), is a peptide pheromone able to induce expression of *comM* outside competence in strain R4846 (see Material and Methods).”

We also provide information about the BIP-1 peptide and details of the construction of the R4846 strain in the Material and Methods section.

19. The reference list does not always use italics for the scientific names of bacteria.

Corrected, thank you.

Reviewer #3 (Remarks to the Author):

We thank the reviewer for their time dedicated to review our manuscript and for their helpful feedback.

Reviewer #4 (Remarks to the Author):

Review for Juillot et al. 'Transient inhibition of cell division in competent pneumococcal cells results from deceleration of the septal peptidoglycan complex'

This submitted manuscript provides significant insights into the mechanisms of how a competence-induced protein ComM interferes with cell division. This manuscript uses state-of-the-art microscopy techniques to study the movements of PG synthesis machinery in relationship to competence development. In combination with genetic screens and biochemical characterization, the authors present a model for the regulation of septal PG synthesis by ComM via DivIB.

A previous report showed that ComM, a membrane protein induced during pneumococcal competence, inhibits cell division. This current paper demonstrates that the induced ComM moves together with the FtsW:PBP2x septal synthesis complex, and reduces the speed of this complex. The interference with PG synthesis is specific to the septal complex, while the movement of the peripheral complex is not affected. A genome-wide Y2H screen identified DivIB as a potential interactor with ComM, which was subsequently confirmed with split-luciferase complementation assays and co-immunoprecipitation experiments. The findings are interesting in terms of general mechanisms of PG machinery movements in response to internal or external stress.

The manuscript is carefully and clearly written. The rationales are clearly stated and the quality of the data is high. I appreciate the detailed description of strain constructions and the different assays. The references are complete and appropriate. Their lines of work will further the use of examination of molecule movements as an assay for physiological phenomenon.

We thank the reviewer for their careful reading of our manuscript, constructive criticisms and helpful suggestions which really helped us improve the manuscript.

Major points

1. Include DivIB-mNG in the analysis performed in extended data fig. 1A. This will also answer the question whether the addition of CSP changes DivIB amounts. Line 418-421 'FtsW and PBP2x

protein levels were comparable in competent and non-competent cells (Extended Data Fig. 1A), indicating that ComM does not affect the amount of active septal PG synthase complexes. However, this raises the intriguing possibility that ComM might specifically degrade DivIB as substrate.' The authors therefore must have questioned if DivIB amounts change in response to CSP addition. DivIB-mNG was detected in the membrane lysate fraction in Fig.6C. If DivIB is below the detection level using the whole cell extract, membrane lysate fractions may be necessary.

The reviewer makes a valid point. We added mNeonGreen-DivIB in the western blot analysis of the fluorescent protein fusions used in this study (Supplementary Fig. 1A). This analysis indicated that the level of DivIB is similar in competent and non-competent cells. In Supplementary Fig. 1A, whole cell extracts were prepared 15 minutes after competence induction. To make sure that the amount of DivIB does not change at earlier or later time point during the course of competence, we analyzed the amount of the mNeonGreen-DivIB fusion at different times after competence induction in otherwise WT or $\Delta comM$ mutant backgrounds (strains R4869 and R5384, respectively). Western blot analysis confirmed that the level of DivIB remains stable during competence in both backgrounds (Supplementary Fig. 13). It is thus very unlikely that DivIB is a degradation substrate of ComM. We added this finding in the text (lines 379-389), new Supplementary Fig. 13, and rephrased the discussion section (lines 422-429).

2. Addition of a summary table or graph to show the velocities of different molecules under different conditions. Include the full name of the fusion protein (e.g. mNG-comM, FtsZ-mNG etc) and conditions (competent cells, noncompetent cell; methicillin addition; inducer addition etc). This will be very useful for cross-referencing.

We thank the reviewer for this suggestion. We have included this information as the new Supplementary Table 1 in the revised manuscript.

3. Co-immunoprecipitation experiment in Fig. 6C: This figure should include negative control strains that do not have mNG tag (for sample, untagged PBP2x FtsW-ALFA strain and untagged DivIB ALFA-ComM strain). This is to control for nonspecific binding of proteins to the mNeongreen trap.

The reviewer raises an important point and we thank them for pushing us to do these controls. We have added a new strain producing the ALFA-ComM fusion (strain R5298) as a negative control in our *in vivo* mNeonGreen co-immunoprecipitation experiments to test the interaction between mNeonGreen-DivIB and ALFA-ComM (strain R5300). Absence of ALFA ComM in the R5298 immunoprecipitates confirmed the presence of DivIB and ComM in the same complex.

In these co-immunoprecipitation experiment, we also originally used the interaction between PBP2x and FtsW as a positive control (strain R5303 containing both mNeonGreen-PBP2x and FtsW-ALFA). As suggested by the reviewer, we added a strain producing the ALFA-FtsW fusion in an untagged PBP2x strain (strain R5299). During the course of the revision, we found however that the FtsW-ALFA fusion was detected in the negative control immunoprecipitates (see Figure below, panel A). We therefore repeated the immunoprecipitation experiments with strains containing either the mNG-PBP2x and FtsW-HaloTag fusions (strain R4745) or the FtsW-mNeonGreen and HaloTag-PBP2x fusions (strain R5386), together with appropriate negative controls (see Figure below, panel B). Note that the growth rate of all these strains were similar to the WT untagged strain, suggesting that all fusions were functional.

Panel B in the figure below shows that regardless of the strains used, we haven't been able to detect an interaction between PBP2x and FtsW in our co-immunoprecipitation conditions (please, see also our response to comment 9 of Reviewers 2 and 3). *In vivo* co-immunoprecipitation of FtsW and PBP2x has previously been achieved using paraformaldehyde crosslinked cultures (Perez et al. 2019, doi:10.1073/pnas.1816018116), which may stabilize the PBP2x:FtsW complex. In all, we consider that these results further support our conclusion that the interaction detected between ComM and DivIB in non-crosslinked conditions is valid.

mNG-DivIB label should be DivIB-mNG.

We apologize for the mistake and thank the reviewer for spotting it. mNeonGreen is actually fused to the N-ter of DivIB, so it was actually correctly labelled in Fig. 6c (mNG-DivIB) but incorrectly mentioned at several places in the main text (DivIB-mNeonGreen). We made the correction in the revised manuscript.

4. Include MW bands in the blot. The section of the blot shown should include the area with one MW size above and one MW size below. It will be useful to add a full-size blot in the supplemental figure. Put the labels of the antibody on the side of the panels so that it is clear that left and middle panels are labeled with anti-mNG and anti-ALFA. Label the mNG proteins as 'bait' and the ALFA proteins as 'prey'. Mention the numbers of biological replicates performed.

We have modified Fig. 6c and Fig. S1 according to the recommendations of the reviewer.

Minor points

Figures:

5. Extended Data Fig. 1: Mention in the figure legend the expected size of mNG, untagged protein and the fusion proteins. To demonstrate that the lower band obtained with anti-HT is a nonspecific band, include a lane obtained with a WT sample. Alternatively, use an anti-HaloTag monoclonal antibody (Promega, G921A, 1:1000), which do not give non-specific bands.

We have added the expected size for fluorescent mNeonGreen and HaloTag fusions in the figure legend. We are very grateful to the reviewer for suggesting the use -and providing the reference- of the anti-HaloTag monoclonal antibody.

6. Some of the labeling and lines on the graphs (Fig. 1 H, 1I, 2B, 2C, 2F, 3, 5, 6D and 6G). are very light. Change the light orange, light blue or yellow to a deeper shade.

We have modified all graphs according to the recommendations of the reviewer. Thanks for the suggestion

7. Mention the numbers of biological replicates performed for experiments shown in the figures. For example, mention the number of biological replicates in figure 1H. For figure 1I, are the values for each time point obtained from one experiment or from multiple experiments? The same clarifications apply to Fig. 2C, 2D and others.

For Figure 1i, the data set is the same as in panel 1h. We analyzed a total of 752 trajectories over three independent biological replicates. This information is now in the revised manuscript.

8. For other experiments that show representative results (fig. 1B, 1C, 1E, 1F, 2A etc), provide the number of biological replicates performed for each kind of experiment.

The number of biological replicates was indicated in the first sentences of the figure legends. For clarity, we have now added this information in the figure legend for each figure panel when necessary. Also, please note that in the original manuscript, most of the graph were generated with data from only one of two or three independent replicates. All figures now include the data from all the replicates, thereby attenuating experimental variations.

9. Legend to Figure 2B and 2F, mention that the color scheme in 2F corresponds to the cells with particular Z-ring profile.

Amended as suggested.

10. Figure 5, need to clarify BIP (see comments on Materials and Methods). Label mNG-pbp2x movement on the graph.

We added the following sentence in the legend of Figure 5:

“BIP-1 (noted BIP on the graph for simplicity), is a peptide pheromone able to induce expression of *comM* outside competence in strain R4846 (see Material and Methods).”

We also provide information about the BIP-1 peptide and details of the construction of the R4846 strain in the Material and Methods section.

11. Legend to figure 6 ‘Whole cell extracts prior to immunoprecipitation (L, load)’, is L the whole cell extracts, or is it the membrane extract? Add volume loaded for each lane. Please also add volume used for the buffers used for the membrane preps and resuspension in the materials and methods section.

We thank the reviewer for noticing this mistake. The load corresponds to “detergent-solubilized membranes”. We made the correction in the legend of Figure 6. The volume loaded for each lane and the volume of buffers used for the co-immunoprecipitation experiment are now specified in the Materials and Methods section. We also indicate the “equivalent culture volume” loaded for L, FT and IP in the legend of Figure 6.

Text:

12. Line 74-76: ‘The model that has emerged is that FtsZ treadmilling is not rate-limiting for septum constriction and does not control the processive movement of active septal PG synthases as originally suggested.’ Put in references after ‘septal PG synthases’ and put in reference after ‘originally suggested’.

We have now corrected this sentence with appropriate references. Please see our response to comment 8 of Reviewer 1.

13. Line 97-98: 'ComM localizes at midcell and does not affect assembly and localization of divisome components, suggesting that it transiently inhibits the active process of constriction.' Provide a reference for this sentence or mention that this is shown in this study.

We added the following reference: Bergé et al. 2017 (doi:10.1038/s41467-017-01716-9).

14. Line 130: mention that microscopy was performed at 37°C because other groups perform microscopy at different temperatures. Add a reference for HILO microscopy.

The text has been modified as suggested. We added the following reference for HILO microscopy: Tokunaga et al. 2008 (doi:10.1038/nmeth1171).

15. Line 177: perhaps add 'but not the components of the peripheral PG synthase complex' to emphasize the effect is specific to the septal PG synthase complex.

Nature Communications recommends to keep subheadings under 60 characters, but we followed the suggestion of the reviewer here.

16. Line 292: PBP2x displayed the same average speed than in wild-type cells in the absence...Change 'than' to 'as'.

Corrected.

17. Line 323, also change 'than' to 'as'

Corrected.

18. Lines 331 to 332: MurA is mentioned in this manuscript only once here. Give a one-sentence background on MurA and MurZ. 'MurA and MurZ are the two homologues of the MurA family in pneumococcus that catalyze the first committed step of PG synthesis.' Give a spr # after MurA because the identities of the two homologues can be confusing.

Strain R5323 containing the MurA (Spr1781) fusion to the ALFA-tag was originally used as a negative control in the co-immunoprecipitation experiments (Fig. 6C). As suggested by this reviewer (see point 3 above), we have replaced this negative control by strains R5298 and R4744, producing the ALFA-ComM and HT-PBP2x fusions, respectively. To avoid confusion, we decided to remove the MurA-ALFA construct from the revised manuscript.

19. Different sections of Discussion: put in figure # so readers can refer to the figures. For example, line 371 'Our findings indicate that the transient inhibition of cell division that occurs during competence in *S. pneumoniae* results from ComM-dependent deceleration of FtsW:PBP2x (Fig. 3A to 3C), etc.

We have amended the discussion as suggested. Thank you.

20. Line 407, *S. pneumonia* should be *S. pneumoniae*.

Corrected.

21. Line 413-414, reference also 43, Straume et al, 2017.

Thank you for pointing this omission out. The reference is now added.

22. Line 457, no supplementary for Table 1 and 2.

Corrected.

23. Line 474, the references 86 and 87 did not use the term BIP-1. Please specify.

We apologize for the confusion. Formal references 86 and 87 correspond to the discovery of the *blp* cluster (bacteriocin-like peptide) producing bacteriocins and their immunity proteins under the control

of a peptide pheromone encoded by *BlpC*. The term BIP-1 was later introduced by Knutsen and colleagues (doi:10.1128/JB.186.10.3078-3085.2004). We have replaced references 86 and 87 by the publication by Knutsen et al. 2004.

24. Line 527, construction of R4846 (Δ comC, mNeonGreen-pbp2x, CEP_R-comM (kan)) is missing

The construction of strains R4845 and R4846 were mentioned with a footnote at the bottom of Table 1. For better clarity, we added the details of these two constructions in the Material and Methods section. We also provide information about the CEP_R inducible expression platform and the CEP_R-comM (kan) construct (see point 28 below).

25. Line 678 to 680, include the volume of lysate added to each lane in the figure legend or in the Materials and Methods. 'Gels were activated by ultraviolet (UV) exposure for 45 s using a Bio-Rad ChemiDoc MP imager to visualize and estimate total protein per lane.' What did you do with the estimation? Did you try to do quantitation and normalize the relative amounts of Pbp2x etc with or without CSP addition?

For immunoblot analysis (Supplementary Figure 1), we loaded 10 μ l of whole cell extracts resulting from cultures treated with CSP (corresponding to 600 ml of cells) and equivalent volumes of non treated cells normalized by OD550. This information is now included in the revised Material and Methods section.

To visualize and control the total amount of proteins loaded, we used the stain free technology. Stain-free total protein measurement served as loading control to quantify the relative levels of fluorescent proteins, for competent versus non-competent cultures. The corresponding values are now shown on Supplementary Figure 1.

26. Line 710, spell out TIRF and HILO

Modified as suggested.

27. Lines 852 to 872, include volume for each step.

Amended as suggested.

28. Table 1: what is CEP_R in R4846 Δ comC, mNeonGreen-pbp2x, CEP_R-comM (kan)?

CEP_R is a BIP-1 inducible expression platform. The CEP_R-comM (kan) construct allows expression of *comM* outside competence under the control of the P_R promoter inducible by the BIP-1 peptide (Knutsen et al., 2004, doi:10.1128/JB.186.10.3078-3085.2004), at the CEP (Chromosomal Expression Platform) locus (Guiral et al. 2006, doi:10.1099/mic.0.28433-0). The BIP-1 inducible CEP_R expression platform was previously described in Johnston et al. 2016 (doi:10.1111/mmi.13360). The BIP-1 peptide (Bacteriocin-Inducing Peptide) controls an autoinduction system regulating the expression of bacteriocins in *S. pneumoniae* (de Saizieu et al., 2000, doi: 10.1128/JB.182.17.4696-4703.2000; Reichmann and Hakenbeck, 2000, doi:10.1111/j.1574-6968.2000.tb09291.x). This peptide induces the *PblpIJK* promoter, renamed P_R for simplicity as it is controlled by the response regulator BlpR.

We have changed CEP_R for CEP_R, and added the construction of strain R4846, including the description of the CEP_R inducible expression platform and the CEP_R-comM (kan) construct, in the Materials and Methods section.

29. R5323 ((mNeonGreen-ComM, MurA-ALFA) is not listed. Strain construction of this strain should be included in Materials and Methods.

We thank the reviewer for noticing this omission. This strain, which was originally used as a negative control in the co-immunoprecipitation experiments (Fig. 6C), is now replaced with strains R5298 and

R4744 (see points 3 and 18 above). The construction of strains R5298 and R5299 are added in the Materials and methods section and in Supplementary Table 1.

Reviewer #4 (Remarks on code availability):

I am not familiar with codes.

POINT-BY-POINT RESPONSES TO REVIEWERS COMMENTS

[manuscript # NCOMMS-24-65120-A, by Juillot *et al.*]

Reviewer #1 (Remarks to the Author):

The authors have addressed all my concerns, and I strongly believe this manuscript should be published. All authors on this work should be congratulated for this great paper.

Thanks! We thank the reviewer for their recommendation and enthusiastic comment.

Reviewer #2 (Remarks to the Author):

The authors have done a thorough job answering and rebutting my comments. The revised manuscript is an important contribution.

Thanks! We thank the reviewer for acknowledging our efforts answering and rebutting their comments and for the positive feedback.

Reviewer #3 (Remarks to the Author):

We thank the reviewer for taking the time to co-review our work and for facilitating training of early career researchers.

Reviewer #4 (Remarks to the Author):

I appreciate very much the detailed revision the authors assembled for the manuscript. I agree with all the authors' changes, and really appreciate the explanations. I look forward to seeing the final version. It is an important contribution to the area.

Thanks! We thank the reviewer for acknowledging our efforts in assembling the revised version of the manuscript, and for her interest and thorough review. We respond to the reviewer's final comments below.

For the section on immunoprecipitation, I appreciate that the equivalence of the volume of the original culture is specified. I always wondered about the relative amounts of pull down in other publications.

Thank you

I just have a few minor comments and suggestions

Figure 6: add 'bait' under mNG-DivIB, and 'Prey' under ALFA-ComM. It also will be useful if R5300 (mNG-DivIB ALFA-Com) and R5298(ALFA-ComM) are added to the figure. It will make it easier to follow the figure.

Corrected as suggested. Thank you.

Supplementary Figure 3: Please add whether the cultures were treated with csp for a certain duration.

We modified the figure legend to specify this, as follows:

Strains R4599 (FtsZ-mNG), R4743 (mNG-PBP2x), R4728 (FtsW-mNG), WT gfp-pbp2b (GFP-PBP2b), R4867 (RodA-mNG), R4869 (mNG-DivIB), R4744 (HT-PBP2x) and R5297 (mNG-ComM, RodA-HT) were grown in C+Y medium to $OD_{550nm} \sim 0.1$ (without CSP induction) and analyzed by epifluorescence microscopy. Strain R4601 (mNG-ComM) was induced to develop competence with CSP at $OD_{550nm} \sim 0.1$ for 15 minutes before imaging.

Supplementary Figure 11: please also add the labels 'bait' and 'prey' to this figure.

Modified as suggested.

Main text, under materials and methods

Whole-cell extracts preparation and immunoblot analysis. Line 707: 'Samples (3 ml) were collected by' ...is different from line 713 to 714 'For immunoblot analysis, we loaded 10 μ l of whole cell extracts resulting from cultures treated with CSP (corresponding to ~600 ml of cells). The cell volumes are different in the two sentences.

We gratefully thank the reviewer for the careful reading of the manuscript and for spotting this mistake! There was indeed an error in the composition of the loading buffer used to prepare samples for immunoblot analyses. We prepared a total of 62.5 μ l of whole cell extract from 3 ml of cells, and 10 μ l of the whole cell extracts (corresponding to 480 μ l of cells) were loaded on gels. We modified the text to make this correction (Lines 694-699).

Line 919: mixed with mNeonGreen-Trap Magnetic Agarose beads (Chromotek) and rotated for 6 h at 4 °C. Can the incubation be longer than 6 h? It will take all day to get the preparation to this step. I assume it will be easier if the experimenter can go home and come back the next day to finish the preparation. Please give a range of time (6 to 12 or 6 to 18 , for example) that would work for this method.

We tried 4 h and 6 h incubation time and found that we obtained maximal immunodepletion of mNeonGreen-DivIB from the lysate after 6 h (virtually no mNeonGreen-DivIB detected in the Flow Through). We do not know if the complex would remain stable for longer times. However, we found that we could introduce a pause in the experiment by keeping the crude membranes at -80°C up to a few weeks before solubilization and incubation with the agarose beads. We have now added this information (Line 897).

A minor comment: supplementary figure 1a, anti-HT: As suggested to the authors, the expected sizes of the fusion proteins were added in the figure legend. The authors also

switched to a commercial antibody that does not give nonspecific band. However, the figure still has one issue. RodA-HT MW is expected to be 81 kD. but the band shown is about 60 kD. A negative control (WT) with a strain with no HT tag will clarify that the band is not non-specific. However, this point is shown by the absence of extra bands in the ht-pbp2x samples. Since anti-rodA is not available, it is not possible to determine whether rodA-HT is degraded. It is not necessary to add a non-tag strain in this case because of the pbp2x samples. However, I want to re-emphasize the need to validate all antibodies (commercial or home-made) for their specificity.

Ho-Ching Tiffany Tsui

We fully agree with the reviewer and we apologize for this omission.

We have no explanation for the aberrant migration of the RodA-HT fusion other than RodA is a multispinning membrane protein, which frequently migrate differently from their calculated molecular weight. This phenomenon was inherent to RodA as we also found that the RodA-mNG fusion, which has an expected size of 72 kD, migrates significantly lower than the FtsZ-mNG fusion (expected size of 74 kD). It is well known that multispinning membrane proteins often migrate at an apparent molecular weight different from their real size on SDS-PAGE gels, either by incomplete denaturation (SDS may not fully unfold the protein, resulting in a more compact structure that migrates faster and thus appears smaller), unusual conformations or reduced SDS binding